# Less is More: Geometric Unlearning for LLMs with Minimal Data Disclosure

**Chenchen Tan** [1]  **Xinghao Li** [1]  **Shujie Cui** [1]  **Youyang Qu** [2][3]  **Cunjian Chen** [1]  **Longxiang Gao** [2][3]

## Abstract

As large language models (LLMs) are increasingly deployed in real-world systems, they must support post-hoc removal of specific content to meet privacy and governance requirements. This motivates selective unlearning, which suppresses information about a particular entity or topic while preserving the LLM's general utility. However, most existing LLM unlearning methods require access to the original training corpus and rely on output-level refusal tuning or broad gradient updates, creating a tension among unlearning strength, non-target preservation, and data availability. We propose Geometric Unlearning (GU), an approach that operates directly on the model's prompt-conditioned hidden states without access to the original training corpus. Specifically, GU distills a compact, low-rank safe-behavior subspace from a small set of safe reference prompts and uses lightweight anchor-in-context synthetic prompts to trigger localized, projection-based alignment of hidden representations to this safe subspace. A teacher-distillation regularizer on synthetic non-target anchors further reduces collateral drift. Across privacy-oriented unlearning benchmarks (ToFU and UnlearnPII), GU achieves strong target suppression with minimal impact on non-target performance, demonstrating that effective unlearning can be achieved with minimal synthetic data.

[1]Faculty of Information Technology, Monash University, Clayton, Victoria, Australia. [2]Key Laboratory of Computing Power Network and Information Security, Ministry of Education, Shandong Computer Science Center, Qilu University of Technology (Shandong Academy of Sciences), Jinan, China. [3]Shandong Provincial Key Laboratory of Computing Power Internet and Service Computing, Shandong Fundamental Research Center for Computer Science, Jinan, China. Correspondence to: Longxiang Gao <gaolx@sdas.org>, Youyang Qu <quyy@sdas.org>.

*Proceedings of the $43^{rd}$ International Conference on Machine Learning*, Seoul, South Korea. PMLR 306, 2026. Copyright 2026 by the author(s).

## 1. Introduction

As LLMs are integrated into more real-world workflows, selective unlearning is becoming a practical requirement driven by evolving user expectations and increasingly strict regulatory and governance constraints (Liu et al., 2025; Fan et al., 2025a; Muresanu et al., 2025). *Selective unlearning* aims to suppress knowledge about a particular person, topic, or dataset while preserving the model's remaining capabilities (Liu et al., 2024b; Wan et al., 2025). Existing work typically pursues this goal either via prompt engineering and inference-time interventions (*e.g.,* eliciting refusals or unmatched responses on target prompts) (Liu et al., 2024a; Pawelczyk et al., 2024) or via parameter updates using gradient-based procedures over designated unlearning and retention datasets (Jang et al., 2023; Wang et al., 2025). However, these strategies typically rely on access to the original training corpus (Liu et al., 2025; Wang et al., 2025), which not only limits applicability in deployed settings but can also re-expose sensitive data during unlearning, creating a central challenge for practical selective unlearning.

This motivates the study of original-corpus-free selective unlearning: removing target knowledge without requiring access to the pretraining corpus, which is often unavailable in practice due to privacy, licensing, or governance constraints. Under this constraint, the standard data-driven strategy for unlearning becomes problematic. In particular, most gradient-based pipelines (Chen & Yang, 2023; Ji et al., 2024; Tan et al., 2025) rely on (i) target-related training evidence to identify what to remove and (ii) a large, distribution-matched retain stream to prevent collateral degradation; both are tightly coupled to the original pretraining distribution and are difficult to obtain without corpus access. This motivates us to look for data-efficient unlearning levers inside the model, where compact internal signals can be estimated and controlled using lightweight prompts rather than corpus-scale supervision (Spohn et al., 2025).

Recent evidence indicates that, after processing the prompt and before generation, LLM hidden representations already carry linearly decodable signals that are predictive of response-level properties, including structural attributes (*e.g.,* expected length and reasoning-step structure), salient content choices (*e.g.,* the eventual multiple-choice answer), and confidence-related proxies (Dong et al., 2025b; Pal et al.,

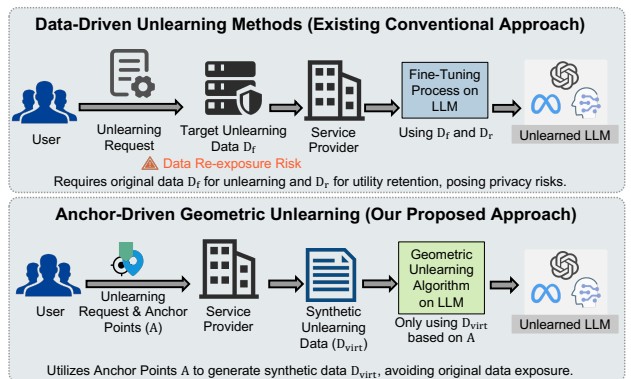

*Figure 1.* Conventional data-driven unlearning vs. our original-corpus-free unlearning (GU). *Top:* Standard unlearning pipelines fine-tune an LLM using target unlearning data $D_f$ and retention data $D_r$, which can re-expose original data and pose privacy risks. *Bottom:* Our approach uses only user-provided anchor points $A$ to generate synthetic unlearning data $D_{virt}$, and applies Geometric Unlearning on the LLM using $D_{virt}$, avoiding access to the original training corpus while enabling targeted unlearning.

2023; Ashok & May, 2025). This indicates that a substantial portion of the model's response plan is instantiated in the prompt state, making it a natural and data-efficient intervention point. If we can reshape this state on target prompts, we can redirect downstream generation without relying on corpus-scale supervision. At the same time, a growing body of work suggests that many high-level semantic and behavioral factors are geometrically structured in representation space and are often well-approximated by low-dimensional linear subspaces (or a small number of salient directions) that simple probes can recover (Saglam et al., 2025; Park et al., 2024; Hernandez & Andreas, 2021; Zhao et al., 2025). In many settings, targeted interventions that add or subtract small components along identified activation-space directions can causally steer model behavior (*e.g.,* reducing toxicity or inducing persona shifts), and analogous effects can be achieved in weight space via task-vector arithmetic (Turner et al., 2023).

Taken together, these observations suggest a different strategy for unlearning as shown in Fig. 1: rather than following the conventional methods (Wang et al., 2025; Cha et al., 2025) pushing the model's output toward a refusal template or finding substitutes for the target, we directly modify the prompt-conditioned hidden representation (Dong et al., 2025b) so that the planned trajectory for the target topic becomes uncertainty-oriented. Specifically, when a prompt is recognized as about a target entity, we impose supervision on the corresponding prompt representation so that the model's implicit planning variables encode uncertainty, *i.e.,* a state consistent with "I'm not sure", rather than a content-generating trajectory. As a result, at inference time, the same class of target prompts induces a planning state that already treats the target knowledge as unavailable.

We therefore introduce **Geometric Unlearning (GU)**, a representation-level approach that combines prompt-conditioned supervision with explicit geometric operations in hidden space. Our method proceeds in three steps. First, we construct a safe-behavior subspace by collecting a small set of prompts that reliably elicit refusal-/uncertain-style behavior and extracting the corresponding prompt-conditioned hidden representations. Second, we synthesize target-entity prompt instances by inserting a target anchor into diverse contextual templates (covering different attributes and contexts) and record the model's hidden states in a short anchor-local window of tokens. Third, we train the model to align the hidden states of these target prompts with the safe-behavior subspace: specifically, for each selected layer, we decompose the centered hidden state into a component within the safe-behavior subspace and a residual component in its orthogonal complement, then optimize two complementary terms: a centroid-pull loss that moves the target hidden state toward the safe-reference mean, and a fold-back confinement loss that penalizes normalized residual energy outside the safe-behavior subspace. To preserve non-target behavior, we add localized teacher-alignment losses that keep (a) the window-external states on target prompts and (b) the retain states on non-target prompts close to a frozen teacher model. Compared to output-space unlearning, GU operates directly on the directional structure of hidden states that encodes topic-specific planning. Extensive experimental results on ToFU (Maini et al., 2024) and UnlearnPII (Parii et al., 2025) benchmarks demonstrate that GU achieves strong unlearning effectiveness and limited utility degradation without accessing the original training corpus. The code is released at `https://github.com/CCT-sys/GU`.

## 2. Related Works

**LLM Unlearning.** LLM unlearning aims to remove a model's reliance on specific training signals, such as personal data, sensitive attributes, or high-risk knowledge. This goal is motivated by regulatory and governance requirements, including the GDPR "right to be forgotten" (Voigt & Von dem Bussche, 2017). Recent work has moved beyond coarse example-level removal and studies selective unlearning targets that better match deployment needs. These targets include entity-level removal (*e.g.,* forgetting a specific person profile) and topic- or capability-level suppression (Ma et al., 2025; Maini et al., 2024; Wan et al., 2025; Li et al., 2024). Most methods rely on training-time parameter updates. They optimize an explicit unlearning objective on a target unlearning dataset. Their designs include discouraging target continuations using negative or unlikelihood-style losses, pushing the model toward uncertainty or refusal, or directly reducing the likelihood of target answers (Liu et al., 2025; Jang et al., 2023; Yao et al., 2024). Another type of unlearning adopts an alignment or preference-optimization

view. It encodes "not producing target knowledge" as a preference constraint (*e.g.,* negative preferences). This often improves optimization stability and reduces severe quality regressions compared with likelihood suppression (Zhang et al., 2024; Fan et al., 2025b). Other approaches intervene at the representation level. They modify activations or directions linked to the target knowledge. Typical operations include projection, direction removal, and layer-local rewrites (Li et al., 2024; Shen et al., 2025; Dang et al., 2025).

**Progress and the Overlooked Data-Dependence Premise.** The latest studies illustrate that, while these approaches can achieve strong unlearning on benchmark metrics, they often introduce two practical trade-offs: degraded non-target utility and vulnerability to recovery after further fine-tuning. To reduce utility loss, many methods add model-utility maintenance strategies to the unlearning framework. These include matching the pre-unlearning model distribution, aligning on retain behavior, and using calibration losses to stabilize generation quality (Tan et al., 2025; Chen & Yang, 2023; Yao et al., 2024). On the other hand, to improve robustness under subsequent fine-tuning, recent work studies relearning-resilient objectives and robustness-based perspectives that make unlearning harder to reverse (Fan et al., 2025a).

However, these LLM unlearning methods still assume access to the target unlearning dataset. Many of them also require retaining data for optimization or calibration. This assumption is often unrealistic in deployed settings. It can also increase privacy risk, because sensitive examples must be handled again during unlearning. This creates an additional exposure surface and complicates compliance claims (Basaran et al., 2025; Ahmed et al., 2025). We therefore treat original-corpus-free LLM selective unlearning as a primary objective. Our goal is controllable unlearning without using any real training samples, while preserving non-target utility and robustness.

## 3. Preliminaries

**Prompt-Conditioned Planning Representations.** Let $x_{1:T}$ be a tokenized prompt and $h_{\ell,t} \in \mathbb{R}^H$ denote the hidden state at layer $\ell$ and position $t$. $h_\ell^{\mathrm{pr}} := h_{\ell,T}$ denotes the prompt-conditioned representation of the prompt. Recent studies show that these prompt-conditioned hidden states already encode global properties of the entire upcoming response, such as topic, style, structure, and length, in a way that can be predicted by simple probes (Dong et al., 2025b). In what follows, we treat $h_\ell^{\mathrm{pr}}$ as a compact summary of the model's planned behavior and use it as a supervised target for steering the planning of specific topics.

**Low-rank Topic Subspaces and Projection Decomposition.** A useful geometric prior for representation-level in-

terventions is that, for many specific semantic or behavioral attributes, the corresponding signal in transformer hidden states is not spread uniformly across all coordinates, but is instead well-approximated by a small number of linear directions (*i.e.,* a low-rank subspace) that is detectable by simple linear readouts (Arditi et al., 2024; Heo et al., 2025; Hernandez et al., 2024). Motivated by this, we model a target topic $z$ as inducing a $k$-dimensional topic subspace at layer $\ell$. Let $\mathcal{H}_\ell(z) = \{h_\ell^{\mathrm{pr}}(x) : x \text{ refers to } z\}$ be prompt-conditioned states from topic-bearing prompts. We estimate an (approximately) orthonormal basis $V_\ell \in \mathbb{R}^{H \times k}$ by rank-$k$ PCA on mean-centered activations from $\mathcal{H}_\ell(z)$, and define the projector

$$P_\ell := V_\ell V_\ell^\top, \qquad P_\ell^\perp := I - P_\ell. \tag{1}$$

This leads to the decomposition $h_\ell^{\mathrm{pr}} = P_\ell h_\ell^{\mathrm{pr}} + P_\ell^\perp h_\ell^{\mathrm{pr}}$. We treat $P_\ell h_\ell^{\mathrm{pr}}$ as topic-sensitive planning coordinates and $P_\ell^\perp h_\ell^{\mathrm{pr}}$ as topic-agnostic content to be preserved, enabling selective unlearning via interventions confined to the topic subspace.

**Causal Controllability via Activation Intervention.** Beyond linear decodability, many attribute directions are causally controllable: intervening on intermediate activations along a learned direction can reliably steer downstream generation. This has been demonstrated by activation engineering and activation addition methods (Turner et al., 2023; Heo et al., 2025; Arditi et al., 2024), as well as more specialized concept-editing and conditional steering frameworks (Marshall et al., 2024; Lee et al., 2025). These results motivate operating directly on prompt-conditioned representations: by modifying $h_\ell^{\mathrm{pr}}$ within the estimated entity subspace while largely preserving the orthogonal complement, which can redirect the model's planned behavior for the target topic.

## 4. Method: Geometric Unlearning

Geometric Unlearning performs selective unlearning by editing prompt-conditioned planning representations. As shown in Fig. 2, the method consists of three core unlearning stages and one stabilization component. We first build a low-rank safe-behavior subspace from reference prompts that elicit the desired safe behavior for the target (refusal or unlearned-style responses). Next, we construct an anchor-conditioned virtual prompt set by placing the anchor into diverse contexts and attributes (*e.g.,* roles, attributes, intents, and surrounding topics), and then run the model on these prompts to collect the associated prompt-conditioned hidden states across selected layers. We then fine-tune the model so that prompt-conditioned states induced by these virtual prompts are aligned to the safe-behavior subspace. Finally, apply retain-side stabilization to limit non-target drift.

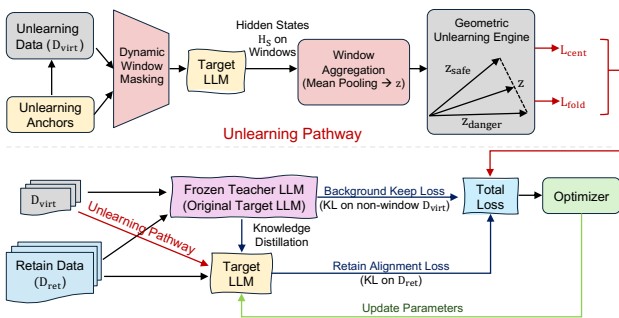

*Figure 2.* Overview of the proposed unlearning framework. The framework is structured into two parallel pathways to balance unlearning and preservation. The top Unlearning Pathway focuses on geometric unlearning by processing target synthetic unlearning data ($D_{virt}$) through dynamic window masking. The aggregated hidden states (for topic $z$) are then aligned within the Geometric Unlearning Engine to minimize centroid-pull ($L_{cent}$) and fold-back ($L_{fold}$) losses. The bottom Preservation Pathway maintains model utility by aligning the trainable target LLM with a Frozen Teacher LLM. This is achieved via KL divergence losses on both synthetic retaining data ($D_{ret}$) and the non-masked background regions of the unlearning data ($D_{virt}$).

## 4.1. Safe-Behavior Subspace Construction

We first define a small set of safe reference prompts $\mathcal{D}_{\text{ref}}$ that are constructed to elicit the expected safe behavior, such as refusal templates and forgetting-style answers. For each reference prompt $x \in \mathcal{D}_{\text{ref}}$, we extract a prompt-conditioned hidden state at layer $\ell$, denoted $h_\ell^{\text{ref}}(x) \in \mathbb{R}^H$. For each layer $\ell \in \mathcal{L}$, we collect reference activations $\{h_\ell^{\text{ref}}(x_i)\}_{i=1}^{N_{\text{ref}}}$ and compute their empirical mean,

$$\mu_\ell^{\text{ref}} \;=\; \frac{1}{N_{\text{ref}}} \sum_{i=1}^{N_{\text{ref}}} h_\ell^{\text{ref}}(x_i). \qquad (2)$$

We then form the centered matrix with rows $h_\ell^{\text{ref}}(x_i) - \mu_\ell^{\text{ref}}$, and compute a PCA basis $V_\ell \in \mathbb{R}^{H \times k}$ with orthonormal columns ($V_\ell^\top V_\ell = I_k$). We define the orthogonal projector onto the resulting $k$-dimensional safe-behavior subspace $\text{span}(V_\ell)$:

$$P_\ell \;:=\; V_\ell V_\ell^\top, \qquad P_\ell^\perp \;:=\; I - P_\ell, \qquad (3)$$

where $P_\ell$ projects a vector onto $\text{span}(V_\ell)$ and $P_\ell^\perp$ projects onto its orthogonal complement. All buffers ($\mu_\ell^{\text{ref}}, P_\ell$) are precomputed and frozen for training.

## 4.2. Anchor-Conditioned Virtual Prompts

To avoid requiring a full unlearning dataset, we generate a compact but high-coverage virtual prompt set $\mathcal{D}_{\text{virt}}$ by inserting the anchor into diverse contextual templates (biographical, instructional, comparative, neutral mentions, *etc.*) and varying surrounding attributes (the generated prompt list in Appendix E). Each virtual prompt $x_{1:T}$ contains one

or more anchor occurrences. For clarity, we present the single-anchor case; multiple occurrences are handled by applying the same windowing procedure to each detected anchor $t$. A local window is defined as,

$$\Omega_t = \{t - w_{\text{pre}}, \ldots, t + w_{\text{post}}\} \cap \{1, \ldots, T\}. \quad (4)$$

The resulting set of hit instances is denoted as $\mathcal{D}_{\text{acc}} = \{(x, t)\}$ and applies all prompt-conditioned alignment losses (Section 4.3) only at the hit token and within $\Omega_t$.

## 4.3. Prompt-Conditioned Alignment to Frozen Safe Geometry

This subsection defines the local alignment signal applied at each anchor hit $(x, t) \in \mathcal{D}_{\text{acc}}$. We leverage the frozen safe-behavior geometry precomputed from $\mathcal{D}_{\text{ref}}$. For each hit, we (i) extract a prompt-conditioned representation via target window pooling, (ii) express it as a deviation from the reference mean, and (iii) align it to the frozen geometry using two complementary terms: centroid pull and fold-back confinement.

For each hit $(x, t)$ and layer $\ell$, form a local prompt-conditioned representation by window ($\Omega_t$) mean-pooling:

$$z_\ell(x, t) \;:=\; \frac{1}{|\Omega_t|} \sum_{u \in \Omega_t} h_{\ell, u}(x) \in \mathbb{R}^H. \qquad (5)$$

Then subtract the frozen reference mean to obtain the mean-shifted state,

$$\tilde{z}_\ell(x, t) \;:=\; z_\ell(x, t) - \mu_\ell^{\text{ref}}. \qquad (6)$$

The deviation is reduced to the reference mean by minimizing the squared distance to $\mu_\ell^{\text{ref}}$:

$$L_{\text{cent}} = \mathbb{E}_{(x,t) \in \mathcal{D}_{\text{acc}}} \left[ \frac{1}{|\mathcal{L}|} \sum_{\ell \in \mathcal{L}} \left\| \tilde{z}_\ell(x, t) \right\|_2^2 \right]. \qquad (7)$$

**In- and Out-subspace Decomposition.** Using the frozen projector $P_\ell$ and its complement $P_\ell^\perp$ (Section 4.1), decompose the mean-shifted state as,

$$\tilde{z}_\ell \;=\; \tilde{z}_\ell^{\text{in}} + \tilde{z}_\ell^{\text{out}},$$

$$\tilde{z}_\ell^{\text{in}} := P_\ell \tilde{z}_\ell \text{ (in-subspace)},$$

$$\tilde{z}_\ell^{\text{out}} := P_\ell^\perp \tilde{z}_\ell \text{ (orthogonal residual)}. \qquad (8)$$

However, under finite-step optimization, mean alignment alone does not explicitly control the relative out-of-subspace component of the residual. To further confine representations to the frozen safe-behavior subspace (captured by $P_\ell$), we define a subspace reflection operator $R_\ell$ and reflect $\tilde{z}_\ell$ with respect to this subspace:

$$R_\ell := 2P_\ell - I, \qquad \tilde{z}_\ell^{\text{fold}} := R_\ell \tilde{z}_\ell \text{ (reflected state)}. \quad (9)$$

By construction, $R_\ell$ preserves $\tilde{z}_\ell^{\text{in}}$ and flips $\tilde{z}_\ell^{\text{out}}$, hence

$$\tilde{z}_\ell^{\text{fold}} = \tilde{z}_\ell^{\text{in}} - \tilde{z}_\ell^{\text{out}}. \tag{10}$$

The stabilized cosine dissimilarity is minimized:

$$L_{\text{fold}} = \mathbb{E}_{(x,t)\in\mathcal{D}_{\text{acc}}} \left[ \frac{1}{|\mathcal{L}|} \sum_{\ell\in\mathcal{L}} \left( 1 - \cos_\epsilon(\tilde{z}_\ell, \tilde{z}_\ell^{\text{fold}}) \right) \right], \tag{11}$$

where $\cos_\epsilon(a,b) = \frac{\langle a,b \rangle}{(\|a\|_2+\epsilon)(\|b\|_2+\epsilon)}$. To see the geometric role of this fold-back term, consider the idealized non-stabilized case with $\epsilon = 0$. One can verify that

$$1 - \cos(\tilde{z}_\ell, \tilde{z}_\ell^{\text{fold}}) = 2\frac{\|\tilde{z}_\ell^{\text{out}}\|_2^2}{\|\tilde{z}_\ell\|_2^2}. \tag{12}$$

Thus, in the idealized non-stabilized case, the fold-back term penalizes the normalized out-of-subspace energy (proof in Appendix A). In practice, $\epsilon$ is used only for numerical stability. Therefore, $L_{\text{fold}}$ provides a relative subspace-confinement signal, while $L_{\text{cent}}$ provides an absolute pull toward the safe-reference mean. This method optimizes the two losses jointly to achieve selective unlearning: $\mathcal{L}_{\text{core}} = L_{\text{cent}} + L_{\text{fold}}$.

### 4.4. Retain-side Stabilization

In practice, applying anchor-conditioned alignment alone can induce collateral drift on non-target behavior, especially for names that are lexically similar to the target anchor. We therefore include a stabilization component that (i) preserves the teacher behavior on background tokens of target prompts $\mathcal{D}_{\text{virt}}$ (non-window), and (ii) preserves behavior on a separate synthetic retain pool. The teacher is initialized from the same checkpoint as the student and kept frozen.

**Background Keep on Target Prompts.** For each forget hit $(x,t) \in \mathcal{D}_{\text{acc}}$ with window $\Omega_t$, define a binary mask $m_u = \mathbf{1}[u \notin \Omega_t]$ over prompt positions. At position $u$, the student and teacher output logits $\ell_u(x) \in \mathbb{R}^{|\mathcal{V}|}$ and $\ell_u^{(T)}(x) \in \mathbb{R}^{|\mathcal{V}|}$ over the vocabulary $\mathcal{V}$. Minimizing masked KL divergence on the non-window tokens:

$$L_{\text{bg}} = \mathbb{E}_{(x,t)\in\mathcal{D}_{\text{acc}}} \left[ \frac{1}{\sum_{u=1}^{T-1} m_u} \sum_{u=1}^{T-1} m_u \, \text{KL}\Big( \sigma(\ell_u^{(T)}(x)) \, \| \right.$$
$$\left. \sigma(\ell_u(x)) \Big) \right], \tag{13}$$

where $\sigma(\cdot)$ is the softmax.

**Synthetic Retain Distillation.** We build a synthetic retain pool $\mathcal{D}_{\text{ret}}$ from prompts about non-target fictional names drawn from a retain-name list. The list includes (1) confusable names that token overlap with the target surface form,

and (2) unrelated names that are randomly generated (The generation prompt is illustrated in Appendix E).

For each retain prompt $x \sim \mathcal{D}_{\text{ret}}$ (with $T_x + 1$ tokens), we align the student to the frozen teacher on all next-token prediction positions $u \in \{1, \ldots, T(x)\}$ (*i.e.,* no window mask is applied). We minimize token-wise KL divergence between the predictive distributions:

$$L_{\text{ret}} = \mathbb{E}_{x\sim\mathcal{D}_{\text{ret}}} \left[ \frac{1}{T_x} \sum_{u=1}^{T_x} \text{KL}\Big( \sigma\left(\ell_u^{(T)}(x)\right) \| \sigma(\ell_u(x)) \Big) \right], \tag{14}$$

where $\sigma(\cdot)$ is the softmax.

We combine the two terms into a single stabilization objective: $L_{\text{retain}} = L_{\text{bg}} + L_{\text{ret}}$. The full optimization objective is to combine the retaining stabilization and the core unlearning objective with equal weight:

$$\mathcal{L}_{\text{total}} = \mathcal{L}_{\text{core}} + L_{\text{retain}}. \tag{15}$$

## 5. Experimental Results

To enable a fine-grained comparison against prior unlearning methods, we integrate our approach into the OpenUnlearning evaluation framework (Dorna et al., 2025). We focus our empirical study on two benchmarks, especially for privacy information unlearning, ToFU (Maini et al., 2024) and UnlearnPII (Parii et al., 2025). ToFU targets person-specific factual content and measures both unlearning and utility preservation under controlled forget/retain splits. Unlearn-PII focuses on removing personally identifiable information (PII) and evaluates whether sensitive strings can still be elicited while maintaining general model behavior. The specific experimental setting and evaluation metrics are stated in Appendix B.

### 5.1. Baselines

OpenUnlearning supports a broad set of unlearning algorithms (Dorna et al., 2025). To avoid confounding implementation and evaluation differences, we restrict our comparisons to baselines already integrated into the framework and select a diverse subset that spans the dominant design space: (i) gradient-based baselines (*e.g.,* GA (Jang et al., 2023), GradDiff (Yao et al., 2024)), (ii) preference-optimization-style objectives (*e.g.,* NPO (Zhang et al., 2024), SimNPO (Fan et al., 2025b)), (iii) localized/representation-motivated methods (*e.g.,* RMU (Li et al., 2024)), and a knowledge distillation method (*e.g.,* UN-DIAL (Dong et al., 2025a)).

### 5.2. ToFU Benchmark Evaluation Results

**Main Results.** Table 1 reports the main comparison under the OpenUnlearning evaluation framework, using its standardized metrics (Appendix B.1) to assess both unlearning

*Table 1.* The evaluation results comparison between the proposed unlearning method and baseline methods on the ToFU benchmark for unlearning 10% dataset. The results include unlearning effectiveness and model-utility maintenance comparison. E.S.: extraction strength of target knowledge; F.R.: ROUGE for target unlearning question answer; Privleak: privacy score; M.U.: Model utility, and red text shows the decrease. ↑ means higher is better, ↓ means lower is better, and ≈ 0 means close to zero is better.

| Methods | LLaMA3.2-1B | | | | LLaMA2-7B | | | |
|---|---|---|---|---|---|---|---|---|
| | Unlearning Effectiveness | | | M.U. | Unlearning Effectiveness | | | M.U. |
| | E.S ↓ | F.R. ↓ | Privleak (≈0) | Avg. ↑ | E.S ↓ | F.R. ↓ | Privleak (≈0) | Avg. ↑ |
| GA (Jang et al., 2023) | **0.032** | **0.000** | **-21.38** | 0.000 (-0.598) | **0.027** | 0.023 | -25.84 | 0.000 (-0.623) |
| GradDiff (Yao et al., 2024) | 0.077 | 0.347 | -36.49 | 0.436 (-0.162) | **0.027** | **0.005** | 63.04 | **0.617** (-0.006) |
| NPO (Zhang et al., 2024) | 0.099 | 0.248 | -43.98 | 0.418 (-0.180) | 0.141 | 0.500 | -90.64 | 0.540 (-0.083) |
| SimNPO (Fan et al., 2025b) | 0.054 | 0.398 | -35.34 | 0.423 (-0.175) | 0.110 | 0.432 | -11.10 | 0.553 (-0.070) |
| UN-DIAL (Dong et al., 2025a) | 0.046 | 0.295 | -96.39 | 0.560 (-0.038) | 0.032 | 0.276 | -94.99 | 0.518 (-0.105) |
| RMU (Li et al., 2024) | 0.062 | 0.358 | -68.09 | 0.448 (-0.150) | 0.027 | 0.097 | 57.38 | 0.030 (-0.593) |
| *GU (Ours)* | 0.172 | 0.06 | -22.27 | **0.577** (-0.021) | 0.114 | 0.058 | **-1.496** | 0.598 (-0.025) |

effectiveness and utility preservation. We summarize unlearning by Extraction Strength (E.S. ↓), which measures how extractable the target continuation is from the model, and Forget ROUGE (F.R. ↓), which measures target-answer overlap between the model output and the ground-truth answer. We measure privacy leakage by PrivLeak (≈ 0), a membership-inference risk score based on AUC for distinguishing forget (member) versus holdout (non-member) samples, calibrated against a retrained model (excluding forget data). Finally, we report Model Utility (M.U. ↑), which aggregates retained performance on ToFU's retain split, real-author split, and general knowledge evaluation.

Across both models (LLaMA3.2-1B and LLaMA2-7B), **GU** achieves a strong overall trade-off: it attains competitive unlearning effectiveness while minimizing collateral degradation. In particular, on LLaMA2-7B, GU matches the strongest baselines in terms of suppression of target behavior, while preserving utility at a high level and achieving a Privleak closest to 0 among the compared methods, indicating near-chance membership inference performance and thus minimal residual privacy leakage. A similar pattern holds on LLaMA3.2-1B, where GU maintains low target overlap (F.R.) and stable utility, remaining competitive with the best-performing baselines. Crucially, GU provides these benefits under a substantially weaker data assumption. Whereas conventional gradient-based unlearning pipelines require access to original training evidence for the target ($D_f$) and a large, distribution-matched retain stream ($D_r$) to stabilize non-target behavior, GU does not need to access the original training corpus: it requires only a lightweight anchor to localize the target entity and uses anchor-in-context synthesis plus representation-level geometric supervision to induce an uncertainty planning state. Thus, GU reaches unlearning and utility preservation comparable to the best baselines while eliminating dependence on the original pre-training corpus, directly addressing data re-exposure and deployability constraints.

**Privacy Risk Under Membership Inference Attacks.** Membership inference attacks (MIAs) provide a direct pri-

vacy test for unlearning by measuring how distinguishable forget examples remain from non-training data using only model-derived signals. The ideal unlearning target is to make the forgotten data undistinguished with the data not trained in the target model (MIAs success is close to random guessing, *e.g.,* close to 50%). Fig. 3 reports the privacy risk of MIAs across unlearning methods on two base models (LLaMA3.2-1B and LLaMA2-7B). Following OpenUnlearning, we quantify privacy risk as the deviation from guess-level membership distinguishability, $|AUC - 0.5|$ (lower is better). For each method, we plot three points corresponding to complementary MIAs scoring rules: Min-K, Reference, and Zlib. Across both base models, GU and GA achieve the near-lowest privacy risk among all methods, indicating that MIAs are driven close to chance performance after applying our unlearning method. These results indicate that GU's unlearning mechanism effectively suppresses membership signals and achieves robust resistance to MIAs across multiple attack scores and model scales.

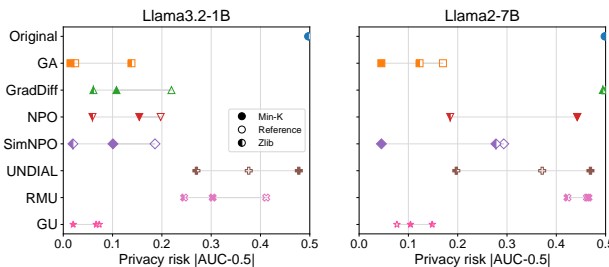

*Figure 3.* Privacy risk of MIAs across unlearning methods for two base models (unlearning 10% benchmark data, *i.e.,* Forget-10) measured by the deviation from chance performance $|AUC - 0.5|$ (lower is better). Each row contains three points computed from different MIAs scoring metrics: Min-K, Reference, and Zlib. For each metric, $AUC$ is the ROC area obtained when using the corresponding attack score to distinguish training members from non-members.

**Unlearned Output Characteristics.** Table 2 summarizes two unlearning-related metrics from OpenUnlearning together with a qualitative categorization of the model's post-unlearning responses on the target unlearning dataset.

*Table 2.* Unlearning output characteristics on the target dataset for LLaMA3.2-1B (Forget-10): Answer Fluency (A.F.; higher indicates less gibberish responses) and Exact Memorization (E.M.; lower indicates less verbatim recall), together with a qualitative categorization of post-unlearning output types.

| Method | A.F.(↑) | E.M. (↓) | Unlearned Output Type |
|---|---|---|---|
| Original | 0.855 | 0.974 | Original |
| GA | 0.285 | 0.000 | Blank |
| GradDiff | 0.725 | 0.646 | Substitute |
| NPO | 0.949 | 0.630 | Substitute |
| SimNPO | 0.896 | 0.566 | Substitute |
| UN-DIAL | 0.592 | 0.515 | Substitute |
| RMU | 0.687 | 0.594 | Substitute |
| GU (Ours) | 0.765 | 0.524 | Refusal/Uncertain |

Answer Fluency (A.F.) measures the extent to which the model's generated answers become nonsensical or uninformative after unlearning, serving as a metric for degraded fluency or "blank" behavior on the target unlearning dataset. Exact Memorization (E.M.) quantifies token-level exact reproduction of the ground-truth continuation, and thus directly captures residual verbatim memorization of the target content.

As shown in Table 2, while several methods substantially reduce E.M. compared to the Original model, they do so with qualitatively different output patterns. The GA baseline drives E.M. to near zero but is associated with Blank or nonsense outputs, indicating that unlearning is achieved primarily by collapsing the response (no meaningful answer and gibberish). In contrast, GradDiff, NPO, SimNPO, UN-DIAL, and RMU predominantly exhibit the Substitute mode, in which the model responds with unrelated content rather than the target answer; this can reduce exact memorization while still producing fluent text, but it may not align with a deliberate safety-compliant unlearning behavior. Notably, GU achieves a competitive reduction in E.M. while producing refusal-/uncertain-style outputs, *i.e.,* the model explicitly expresses uncertainty on the target set. This Refusal-/Uncertain-type behavior represents a more structured unlearning outcome: it suppresses verbatim recall without resorting to empty or gibberish responses and avoids substituting potentially misleading unrelated knowledge.

**Ablation Study.** Fig. 4 characterizes the trade-off induced by the synthetic sample budget when training GU with anchor-triggered prompts. With only 10 samples, both models exhibit relatively high extraction strength, indicating that the virtual prompt set is insufficient to steer prompt-conditioned representations toward the safe planning geometry consistently. Increasing the budget to 30 samples achieves the best balance: extraction strength drops markedly while model utility remains largely stable, indicating that this range provides adequate contextual coverage for anchor-conditioned alignment, and the retaining samples could reduce excessive collateral drift.

Moving from 30 to 40 samples further strengthens unlearning, but the improvement comes with a noticeable utility cost. Under the matched 1:1 forget and retain setting, the additional unlearning pressure at 40 samples is not fully offset by the corresponding retain supervision, and non-target performance degrades. Runtime also increases monotonically with the sample budget for both backbones. Overall, these results motivate using 20 to 30 synthetic samples per anchor as a practical operating point that achieves strong unlearning, preserves utility, and avoids the larger runtime and diminishing returns observed at 40 samples.

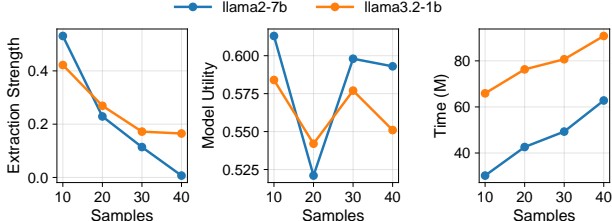

*Figure 4.* Effect of synthetic sample budget on unlearning, retaining, and runtime. We construct 10 to 40 anchor-conditioned synthetic samples for unlearning, paired with an equal number of synthetic retain samples (1:1 forget and retain) for each setting. We report extraction strength (lower is better), model utility (higher is better), and runtime for LLaMA2-7B and LLaMA3.2-1B.

*Table 3.* Effect of the edited target-layer set on unlearning and utility on LLaMA3.2-1B (Forget-10). Epoch to Conv. reports the number of training epochs required to reach unlearning convergence under the same training budget and 1:1 forget and retain sampling.

| Target Layer | E.S.(↓) | M.U. (↑) | Epoch to Conv. |
|---|---|---|---|
| First Layer | 0.498 | 0.588 | 40 |
| First Five Layer | 0.502 | 0.591 | 40 |
| Middle Layer | 0.487 | 0.591 | 40 |
| Last Layer | 0.182 | 0.579 | 22 |
| Last Two Layer | 0.172 | 0.577 | 27 |
| Last Five Layer | 0.165 | 0.568 | 35 |

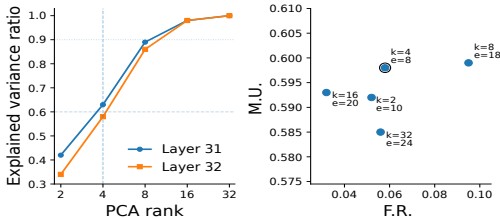

*Figure 5.* Analysis of the safe-behavior subspace and PCA rank selection on the last two layers of LLaMA2-7B. Left: the PCA explained-variance spectrum of safe-reference activations shows that the dominant safe-behavior variation is captured by a compact low-rank subspace. Right: the forgetting-utility trade-off across PCA ranks, where each point is annotated with the corresponding rank $k$ and convergence epoch $e$.

Table 3 reports the effect of the edited layer set on target extraction strength (E.S.), model utility (M.U.), and the epoch

at which unlearning converges. Editing early or middle layers results in nearly identical E.S. values even after 40 epochs, indicating that low- or mid-level representations are not an effective control point for suppressing target knowledge under our prompt-conditioned alignment objective. In contrast, constraining late-layer representations substantially reduces extractability, with E.S. dropping to around 0.17. This is consistent with the intuition that later-layer hidden states are more directly upstream of the final LM head, and therefore have a more immediate influence on the output token distribution. The last five layers' editing results show that increasing the number of aligned late layers strengthens unlearning but slows convergence. Overall, aligning the last one or two layers provides the best trade-off between strong unlearning in LLaMA3.2-1B, preserved utility, and training efficiency.

To test whether the geometric target used by GU is well justified, we examine whether safe-reference activations can be compactly represented by a low-dimensional subspace. As shown in Fig. 5, a rank-4 PCA subspace already captures a majority of the variance on the edited layers, explaining 63% and 58% of the variance on the two layers, respectively. Increasing the rank to 16 captures 98% of the variance on both layers. This result provides empirical support for the core low-rank assumption of GU, showing that the dominant variation of safe behavior is concentrated in a small number of directions rather than being dispersed across the full hidden space. Therefore, a compact PCA subspace can serve as a meaningful geometric target for projection-based alignment. The right sub-figure of Fig. 5 further shows that GU remains effective across a range of PCA ranks, with no consistent benefit from using larger subspaces. Since larger ranks generally require more optimization steps, we use $k = 4$ as a default and efficient operating point.

*Table 4.* Matched source-free comparison using the same synthetic datasets on all baselines.

| Method | E.S. ↓ | F.R. ↓ | PrivLeak ($\approx 0$) | M.U. ↑ |
|--------|--------|--------|------------------------|--------|
| GA | 0.112 | 0.319 | -78.5 | 0.231 |
| GradDiff | 0.513 | 0.615 | -97.6 | 0.561 |
| NPO | 0.627 | 0.751 | -99.4 | 0.594 |
| SimNPO | 0.648 | 0.768 | -99.4 | 0.594 |
| UN-DIAL | 0.701 | 0.818 | -99.5 | 0.600 |
| RMU | 0.696 | 0.797 | -99.5 | 0.592 |
| GU | 0.172 | 0.060 | -22.3 | 0.577 |

**Matched Source-Free Baseline Comparison.** To disentangle the effect of the geometric objective from that of the synthetic-data construction pipeline, we evaluate the baselines under the same source-free setting as GU on LLaMA3.2-1B. Specifically, all methods are trained using the identical synthetic datasets constructed by our pipeline used by GU.

As shown in Table 4, synthetic dataset construction alone

is insufficient to achieve effective source-free unlearning. GA obtains relatively low extraction strength, but also with severe utility degradation. In contrast, GradDiff, NPO, Sim-NPO, UN-DIAL, and RMU preserve higher model utility but fail to suppress the target knowledge effectively, as reflected by their substantially higher E.S. and F.R. GU achieves the best overall unlearning and utility maintaining trade-off under the matched synthetic-data constraint, indicating that its gains cannot be attributed to synthetic data construction alone and that the geometric unlearning objective is essential for effective source-free selective unlearning.

**Computational Efficiency.** As shown in Table 5, GU converges in 31.29 minutes on LLaMA2-7B. Its training time is higher than that of several baselines, but remains comparable to GradDiff and SimNPO and lower than UN-DIAL. This moderate overhead mainly comes from training on a larger synthetic prompt set, which is used to improve contextual coverage around target anchors. The breakdown on the right side of Table 5 shows that the GU-specific preprocessing cost is small: synthetic-data construction takes 12.37 minutes offline, while safe-reference extraction and safe-subspace construction require only 1.59 and 0.34 minutes, respectively. Overall, GU adds little geometry-specific overhead and remains practical in end-to-end runtime.

*Table 5.* Runtime analysis on LLaMA2-7B in the Forget-10 setting.

| Training Time Comparison | | | GU Time Breakdown | |
|--------|------------|-------|-------------------------------|------------|
| Method | Time (min) | Epoch | Stage | Time (min) |
| GA | 8.98 | 7 | Virtual data construction | 12.37 |
| GradDiff | 26.16 | 10 | Safe-reference extraction | 1.59 |
| NPO | 15.23 | 10 | Safe-subspace construction | 0.34 |
| SimNPO | 32.33 | 10 | Training to convergence | 31.29 |
| UN-DIAL | 44.51 | 18 | Total | 45.59 |
| RMU | 2.56 | 6 | | |
| GU | 31.29 | 8 | | |

## 5.3. UnlearnPII Benchmark Evaluation Results

Fig. 6 reports the trade-off between target knowledge suppression and retained model utility on the UnlearnPII benchmark across two model scales (LLaMA3.2-1B, LLaMA3.1-8B), and three forget splits (1%, 5%, 10%). Unlike ToFU, where strong base checkpoints are commonly available, for UnlearnPII, we first train an original model on the provided dataset and then apply unlearning using the official split protocol. Each point in the Fig. 6 corresponds to a method evaluated at convergence; values on the y-axis closer to 0 indicate stronger suppression of target knowledge, while further right on the x-axis indicates better preservation of non-target utility.

Across all splits, GU generally lies in the high-utility region and achieves the competitive target knowledge accuracy

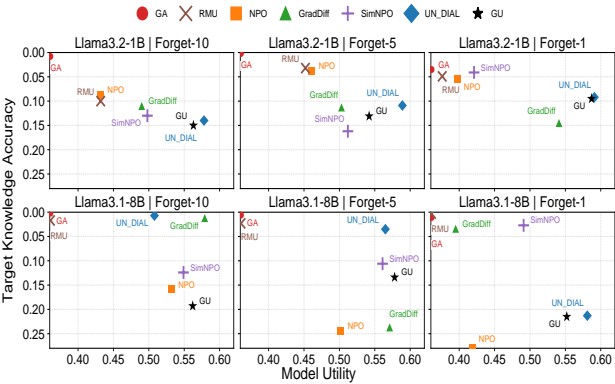

*Figure 6.* Unlearning effectiveness and model utility trade-off across model scales and forget splits on UnlearnPII benchmark. The y-axis reports target knowledge accuracy (close to 0 indicates better unlearning), and the x-axis reports retained model utility (higher indicates better utility preservation).

suppression relative to most baselines, indicating a favorable balance between unlearning and utility maintenance. In particular, methods such as GA and RMU achieve strong suppression but cluster at lower utility, indicating that they obtain unlearning by inducing aggressive behavioral collapse. GU does not always attain the strongest suppression achievable by the most aggressive methods; however, it achieves competitive target suppression while maintaining comparatively high utility.

Importantly, despite the unlearning effectiveness and the utility retention, these GU results are obtained without access to the original pretraining corpus: GU relies only on minimal synthetic prompts constructed from anchors and a small set of safe reference prompts, together with stabilization via synthetic retain anchors. This makes GU particularly suitable for privacy-oriented unlearning settings where training data access is restricted, while still delivering strong practical performance.

**Large-Scale Unlearning.** Fig. 7 shows GU's training trajectories under a large unlearning scale (20% forget split) for two models. For both models, accuracy on the unlearning set decreases smoothly, indicating effective target suppression even when the unlearning dataset is large and diverse. Retaining accuracy exhibits a consistent two-stage pattern. In early epochs, updates are dominated by the anchor-based unlearning objective, and the model begins to drift on non-target behavior, leading to a drop in retaining accuracy. In later epochs, the signal for stabilization becomes stronger: as unlearning-induced drift increases the discrepancy between the target and the frozen teacher on non-target data (background tokens outside the unlearning window and the synthetic retain pool), the distillation KL produces larger corrective gradients that pull the student back toward the teacher on these regions. These dynamics illustrate that the stabilization objective can mitigate non-target drift during

training, enabling retaining accuracy to partially recover while target suppression continues to improve.

Model scale affects both robustness and optimization in this Forget-20 setting. The smaller model shows lower retaining accuracy throughout training, indicating higher susceptibility to interference under strong unlearning pressure. The larger model preserves higher retaining accuracy overall, but requires more epochs to reach convergence, reflecting more gradual optimization under the larger unlearning split. Overall, these dynamics confirm that GU remains effective for large-scale unlearning, while the stabilization component helps restore and maintain utility once non-target drift emerges.

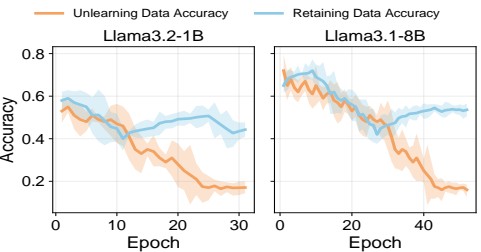

*Figure 7.* Training dynamics under large-scale unlearning (20% forget split) for LLaMA3.2-1B and LLaMA3.1-8B. We track accuracy on the unlearning and retaining sets over training epochs. Shaded bands indicate variability across runs.

# 6. Conclusion

We introduced Geometric Unlearning (GU), a selective unlearning framework that operates on prompt-conditioned hidden states without access to the original training corpus. GU distills a low-rank safe-behavior subspace from a set of safe reference prompts and uses anchor-triggered synthetic prompts to steer target-induced hidden states toward this safe-behavior subspace, while teacher-based stabilization preserves non-target utility. Across privacy-oriented benchmarks, GU achieves strong target suppression with limited utility degradation using only minimal synthetic data supervision, supporting a less-is-more view of unlearning via representation-level control. However, GU is strongest when the training targets are reliably invoked by anchors and when synthetic prompts adequately cover real query contexts during training; performance may degrade for training data constructed with aliases, multilingual variants, or indirect mentions, and distribution shift can leave rare contexts under-covered. Moreover, while GU aligns with standard extractability-based evaluations, it does not certify complete representational erasure under stronger white-box audits. To address these limitations, future work will broaden anchor and contextual coverage, improve prompt generation with adversarial strategies, and strengthen evaluation with representation-level audits; scaling to larger unlearn regimes may further benefit from more targeted stabilization.

## Impact Statement

This paper presents work whose goal is to advance the field of Machine Learning. There are many potential societal consequences of our work, none of which we feel must be specifically highlighted here.

## Acknowledgment

This paper is partially supported by Qilu University of Technology (Shandong Academy of Sciences) Youth Outstanding Talent Program No. 2024QZJH02 and Shandong Provincial University Youth Innovation and Technology Support Program No.2022KJ291. We also acknowledge that this work was supported by Monash eResearch capabilities, including M3.

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

## A. Fold-Back Confinement and Out-of-Subspace Energy

We prove Eq. 12 for the idealized non-stabilized case, *i.e.*, $\epsilon = 0$ and $\tilde{z}_\ell \neq 0$.

**Orthogonal Projector and Subspace Reflection.** Fix a layer $\ell$ and let $V_\ell \in \mathbb{R}^{H \times k}$ contain the top-$k$ PCA directions computed from $\mathcal{D}_{\text{ref}}$, with orthonormal columns $V_\ell^\top V_\ell = I_k$. Define the orthogonal projector onto the PCA subspace

$$P_\ell := V_\ell V_\ell^\top, \qquad P_\ell^\perp := I - P_\ell.$$

Since $V_\ell^\top V_\ell = I_k$, we have

$$P_\ell^\top = (V_\ell V_\ell^\top)^\top = V_\ell V_\ell^\top = P_\ell, \qquad P_\ell^2 = V_\ell (V_\ell^\top V_\ell) V_\ell^\top = V_\ell V_\ell^\top = P_\ell,$$

so $P_\ell$ is an orthogonal projector. Hence every $z \in \mathbb{R}^H$ admits the unique orthogonal decomposition

$$z = z_\| + z_\perp, \quad z_\| := P_\ell z \in \text{span}(V_\ell), \quad z_\perp := P_\ell^\perp z \in \text{span}(V_\ell)^\perp, \quad \langle z_\|, z_\perp \rangle = 0.$$

The reflection across the subspace $\text{span}(V_\ell)$ is defined by keeping the in-subspace component unchanged and flipping the sign of the orthogonal component:

$$R_\ell z := z_\| - z_\perp.$$

Substituting $z_\| = P_\ell z$ and $z_\perp = P_\ell^\perp z = (I - P_\ell)z$ yields

$$R_\ell z = P_\ell z - (I - P_\ell)z = (2P_\ell - I)z,$$

and therefore

$$\boxed{R_\ell = 2P_\ell - I.}$$

**Lemma A.1** (Fold-back confinement suppresses out-of-subspace energy). *Fix layer $\ell$ and let $P_\ell = V_\ell V_\ell^\top$ be the orthogonal projector onto the $k$-dimensional PCA subspace computed from $\mathcal{D}_{\text{ref}}$, and $R_\ell := 2P_\ell - I$ be the corresponding reflection across that subspace. For any mean-shifted vector $\tilde{z}_\ell := z_\ell - \mu_\ell^{\text{ref}}$ (deviation from the reference mean) with $\tilde{z}_\ell \neq 0$, define its in- and out-subspace components*

$$\tilde{z}_{\ell,\|} := P_\ell \tilde{z}_\ell \quad \text{(in-subspace component)}, \qquad \tilde{z}_{\ell,\perp} := (I - P_\ell)\tilde{z}_\ell \quad \text{(orthogonal residual)}.$$

*Let $\tilde{z}_\ell^{\text{fold}} := R_\ell \tilde{z}_\ell$ (reflected state). For the non-stabilized cosine, we have*

$$1 - \cos(\tilde{z}_\ell, \tilde{z}_\ell^{\text{fold}}) = 2 \frac{\|\tilde{z}_{\ell,\perp}\|_2^2}{\|\tilde{z}_\ell\|_2^2}.$$

*Proof.* By the definition of reflection across the projector subspace,

$$\tilde{z}_\ell^{\text{fold}} = R_\ell \tilde{z}_\ell = \tilde{z}_{\ell,\|} - \tilde{z}_{\ell,\perp} \quad \text{(keep } \|, \text{ flip } \perp).$$

Using $\langle \tilde{z}_{\ell,\|}, \tilde{z}_{\ell,\perp} \rangle = 0$ (orthogonality of the projector split),

$$\langle \tilde{z}_\ell, \tilde{z}_\ell^{\text{fold}} \rangle = \langle \tilde{z}_{\ell,\|} + \tilde{z}_{\ell,\perp}, \ \tilde{z}_{\ell,\|} - \tilde{z}_{\ell,\perp} \rangle = \|\tilde{z}_{\ell,\|}\|_2^2 - \|\tilde{z}_{\ell,\perp}\|_2^2.$$

Also, $\|\tilde{z}_\ell\|_2^2 = \|\tilde{z}_{\ell,\|}\|_2^2 + \|\tilde{z}_{\ell,\perp}\|_2^2$ and $\|\tilde{z}_\ell^{\text{fold}}\|_2 = \|\tilde{z}_\ell\|_2$ (reflection is norm-preserving), hence

$$\cos(\tilde{z}_\ell, \tilde{z}_\ell^{\text{fold}}) = \frac{\|\tilde{z}_{\ell,\|}\|_2^2 - \|\tilde{z}_{\ell,\perp}\|_2^2}{\|\tilde{z}_{\ell,\|}\|_2^2 + \|\tilde{z}_{\ell,\perp}\|_2^2} = 1 - 2 \frac{\|\tilde{z}_{\ell,\perp}\|_2^2}{\|\tilde{z}_\ell\|_2^2},$$

which rearranges to the claim for the idealized non-stabilized case. $\square$

This identity shows that, in the idealized non-stabilized reflection view, fold-back confinement corresponds to penalizing the relative residual energy outside the safe-behavior subspace. The stabilized cosine used in the main objective adds $\epsilon$ only for numerical stability. In implementation, we instantiate this confinement principle through projection-direction alignment and direct residual-energy control in the orthogonal complement.

*Table 6.* The baseline hyper-parameters setting in ToFU benchmark evaluation

| Method | Parameters | Learning rate (Forget-01) | Learning rate (Forget-05 & 10) |
|---|---|---|---|
| GA | default | 1e-4 | 1e-5 |
| GradDiff | default | 1e-4 | 1e-5 |
| NPO | default | 1e-4 | 1e-5 |
| SimNPO | beta=0.5; gamma=0.5; delta=0.1; alpha=0.2 | 1e-4 | 1e-5 |
| UN-DIAL | beta=35 | 1e-4 | 1e-4 |
| RMU | alpha=0.5 | 1e-4 | 1e-5 |
| GU | $w_{L_{\mathrm{cent}}}$=0.5; $w_{L_{\mathrm{fold}}}$=0.5; forget_window=8 | 1e-4 | 1e-5 |

*Table 7.* Evaluation metrics used in our experiments, grouped by dimension.

| Dimension | Metrics (reported in this paper) |
|---|---|
| Mem. | **E.M.** Exact Memorization (Tirumala et al., 2022); **E.S.** Extraction Strength (Carlini et al., 2021); **F.R.** ROUGE on forget QA (Maini et al., 2024); **A.F.** Answer Fluency (anti-gibberish) (Mekala et al., 2025) |
| Privacy | **Privleak** Retrain-normalized membership leakage (Shi et al., 2025). |
| Utility | **M.U.** Model Utility (Maini et al., 2024) (aggregate retain-side utility score: include training data excluded forget data and real world knowledge.) |

## B. Experimental Setting

All experimental results were obtained on an NVIDIA GPU workstation equipped with an NVIDIA A100 80GB PCIe accelerator (80 GB HBM2e). Table 6 summarizes our hyperparameter settings. All methods except GU are adopted from the OpenUnlearning framework with their default configurations. Empirically, we find that unlearning becomes harder when the forget set is smaller (*e.g.*, Forget-01), likely due to weaker and noisier optimization signals. Therefore, we use a larger learning rate for Forget-01 than for Forget-05/10 to encourage more effective unlearning while keeping other settings unchanged. We evaluate GU on target models ranging from LLaMA3.2-1B and LLaMA2-7B to LLaMA3.1-8B, demonstrating its effectiveness across different model sizes. The evaluation metrics used in the experimental are shown in Table 7 grouped by three dimension: forget data memorization, privacy, and model utility. The details introduction of each metric shown in below.

### B.1. Evaluation Metrics

**Notation.** Let $f_\theta$ be the evaluated model. For a QA example, let $x$ be the prompt and $y = (y_1, \ldots, y_{|y|})$ be the reference answer tokens. Let $\hat{y} = f_\theta(x)$ denote the generated answer. We write $p_\theta(\cdot \mid \cdot)$ for the model's next-token distribution, and $\mathrm{ROUGE}(\hat{y}, y)$ for a standard ROUGE overlap score.

**Exact Memorization (E.M.).** We measure token-level exact memorization as the fraction of positions where the model's greedy prediction matches the ground-truth continuation:

$$\mathrm{EM}(x, y) = \frac{1}{|y|} \sum_{k=1}^{|y|} \mathbf{1}\Big[\arg\max_v \ p_\theta(v \mid x, y_{<k}) = y_k\Big]. \tag{16}$$

Lower EM indicates less residual verbatim memorization.

**Extraction Strength (E.S.).** Extraction Strength captures how little prefix is needed before the remaining suffix becomes exactly recoverable. Define

$$k^\star(x, y) := \min\Big\{k \in \{0, \ldots, |y|\} \ : \ f_\theta([x, y_{<k}]) = y_{>k}\Big\}, \tag{17}$$

then the extraction strength is

$$\mathrm{ES}(x, y) := 1 - \frac{k^\star(x, y)}{|y|}. \tag{18}$$

Lower ES indicates weaker extractability (better unlearning).

**Forget ROUGE (F.R.).** On forget QA, we compute overlap between the generated answer and the reference answer:

$$\mathrm{FR}(x, y) := \mathrm{ROUGE}(\hat{y}, y).\tag{19}$$

Lower FR indicates the model is less able to reproduce the target answer.

**Answer Fluency (A.F.).** To diagnose collapse into blank or nonsensical outputs, we use a fluency score derived from a gibberish detector. Let $g(\hat{y}) \in [0, 1]$ be the detector's probability that $\hat{y}$ is gibberish; we report

$$\mathrm{AF}(\hat{y}) := 1 - g(\hat{y}).\tag{20}$$

Higher A.F. indicates more fluent, non-gibberish answers.

**Privacy (Membership Inference Risk).** We evaluate privacy leakage via membership inference attacks (MIAs), which attempt to infer whether a sample was seen during training. Following MUSE (Shi et al., 2025), we treat forget examples as members and a disjoint holdout set as non-members, with labels $m(x) = 1$ for $x \in \mathcal{D}_\mathrm{f}$ and $m(x) = 0$ for $x \in \mathcal{D}_\mathrm{h}$. Given a model $f$ and an attack scoring function $s(\cdot; f)$, the attacker predicts membership by thresholding:

$$\hat{m}_\tau(x) := \mathbf{1}[s(x; f) \geq \tau].\tag{21}$$

For any threshold $\tau$, the true positive rate (TPR) and false positive rate (FPR) are

$$\mathrm{TPR}(\tau) := \Pr(\hat{m}_\tau(x) = 1 \mid m(x) = 1), \qquad \mathrm{FPR}(\tau) := \Pr(\hat{m}_\tau(x) = 1 \mid m(x) = 0).\tag{22}$$

To obtain the ROC curve, we vary $\tau$ over all possible thresholds (equivalently, over all distinct score values), and collect the corresponding points $\{(\mathrm{FPR}(\tau), \mathrm{TPR}(\tau))\}_\tau$. The area under this curve is the ROC–AUC:

$$\mathrm{AUC}(f; \mathcal{D}_\mathrm{f}, \mathcal{D}_\mathrm{h}) := \int_0^1 \mathrm{TPR}\big(\mathrm{FPR}^{-1}(u)\big) \, du.\tag{23}$$

Equivalently, AUC admits a threshold-free pairwise form as the probability that a random member receives a higher attack score than a random non-member:

$$\mathrm{AUC}(f; \mathcal{D}_\mathrm{f}, \mathcal{D}_\mathrm{h}) = \frac{1}{|\mathcal{D}_\mathrm{f}||\mathcal{D}_\mathrm{h}|} \sum_{x \in \mathcal{D}_\mathrm{f}} \sum_{x' \in \mathcal{D}_\mathrm{h}} \Big(\mathbf{1}[s(x; f) > s(x'; f)] + \tfrac{1}{2}\mathbf{1}[s(x; f) = s(x'; f)]\Big).\tag{24}$$

We report privacy risk as $\mathrm{Priv} := \big|\mathrm{AUC}(f; \mathcal{D}_\mathrm{f}, \mathcal{D}_\mathrm{h}) - 0.5\big|$ (lower is better), where $\mathrm{AUC} = 0.5$ corresponds to chance-level membership inference.

**PrivLeak.** When a retrained model $f_\mathrm{retrain}$ is available, we additionally report the normalized privacy leakage used in MUSE (Shi et al., 2025):

$$\mathrm{PrivLeak} := \frac{\mathrm{AUC}(f; \mathcal{D}_\mathrm{f}, \mathcal{D}_\mathrm{h}) - \mathrm{AUC}(f_\mathrm{retrain}; \mathcal{D}_\mathrm{f}, \mathcal{D}_\mathrm{h})}{\mathrm{AUC}(f_\mathrm{retrain}; \mathcal{D}_\mathrm{f}, \mathcal{D}_\mathrm{h})}.\tag{25}$$

It is close to zero for good unlearning. Note: The experimental result in each table for this metric is % number.

**Min-K score.** Let $\ell_t(x; f) := -\log p_f(x_t \mid x_{<t})$ be token-level negative log-probabilities. Let $\mathcal{I}_K(x)$ denote indices of the smallest $K = \lceil (k/100)\, T \rceil$ values among $\{\ell_t\}_{t=1}^T$. The Min-K score is

$$s_{\min K}(x; f) := \frac{1}{K} \sum_{t \in \mathcal{I}_K(x)} \ell_t(x; f).\tag{26}$$

Intuitively, member sequences often contain a subset of tokens that are unusually easy for the model, making this statistic discriminative (Shi et al., 2024).

**Zlib-normalized score.** We normalize model perplexity by a compression-based proxy for text entropy. Let $L_\mathrm{zlib}(x)$ denote the compressed length (in bytes) of $x$ under zlib, and define $H_\mathrm{zlib}(x) := L_\mathrm{zlib}(x)/|x|$. Then

$$s_\mathrm{zlib}(x; f) := \frac{\log \mathrm{PPL}_f(x)}{H_\mathrm{zlib}(x)}.\tag{27}$$

This reduces bias from intrinsically low-entropy text (Carlini et al., 2021).

**Reference-model score.** Given a fixed reference model $f_{\text{retrain}}$ (e.g., a retain-trained model), we score examples by a loss gap:

$$s_{\text{retrain}}(x; f, f_{\text{retrain}}) := s_{\text{loss}}(x; f) - s_{\text{loss}}(x; f_{\text{retrain}}). \qquad (28)$$

Using the gap mitigates shared difficulty effects and emphasizes membership-related differences (Dorna et al., 2025).

**Model Utility (M.U.).** We report a single aggregate utility score over the retain-side evaluation suite (as defined by the benchmark protocol). Concretely, M.U. is an aggregate across multiple retain-oriented QA metrics (e.g., probability, ROUGE, and truth-based measures across retain and holdout splits), with higher indicating better non-target preservation.

## C. Additional Evaluation Results

### C.1. Additional Main Results and MIA results

The evaluation results comparison between the proposed unlearning method and baseline methods on the ToFU benchmark for the unlearning 5% and 1% datasets are shown in Table 8 and Table 9, respectively. The privacy risk of membership inference attacks evaluation results comparison across unlearning methods and two base models: LLaMA3.2-1B and LLaMA2-7B (unlearning 1%, 5%, 10% dataset) are shown in Table 10.

*Table 8.* The evaluation results comparison between the proposed unlearning method and baseline methods on the ToFU benchmark for unlearning 5% dataset. The results include unlearning effectiveness and model-utility maintenance comparison. E.S.: extraction strength of target knowledge; F.R.: ROUGE for target unlearning question answer; Privleak: privacy score; M.U.: Model utility, and red text shows the decrease. ↑ means higher is better, ↓ means lower is better, and ≈ 0 means close to zero is better.

| Methods | LLaMA3.2-1B | | | | LLaMA2-7B | | | |
|---|---|---|---|---|---|---|---|---|
| | Unlearning Effectiveness | | | M.U. | Unlearning Effectiveness | | | M.U. |
| | E.S ↓ | F.R. ↓ | Privleak (≈ 0) | Avg. ↑ | E.S ↓ | F.R. ↓ | Privleak (≈0) | Avg. ↑ |
| GA (Jang et al., 2023) | **0.033** | 0.044 | -15.24 | 0.000 (-0.598) | **0.027** | **0.019** | **-9.739** | 0.000 (-0.623) |
| GradDiff (Yao et al., 2024) | 0.087 | 0.374 | -45.97 | 0.457 (-0.141) | 0.207 | 0.446 | -70.72 | 0.568 (-0.055) |
| NPO (Zhang et al., 2024) | 0.091 | 0.313 | -69.39 | 0.447 (-0.151) | 0.246 | 0.534 | -96.72 | 0.510 (-0.113) |
| SimNPO (Fan et al., 2025b) | 0.083 | 0.380 | -49.89 | 0.440 (-0.158) | 0.102 | 0.338 | 16.50 | 0.552 (-0.071) |
| UN-DIAL (Dong et al., 2025a) | 0.051 | 0.267 | -95.42 | 0.569 (-0.029) | 0.045 | 0.317 | -94.75 | **0.570** (-0.053) |
| RMU (Li et al., 2024) | 0.068 | 0.389 | -81.17 | 0.461 (-0.137) | 0.032 | 0.180 | 1.653 | 0.034 (-0.589) |
| GU (Ours) | 0.104 | **0.037** | **1.131** | **0.511** (-0.087) | 0.069 | 0.051 | 28.33 | 0.591 (-0.032) |

*Table 9.* The evaluation results comparison between the proposed unlearning method and baseline methods on the ToFU benchmark for unlearning 1% dataset. The results include unlearning effectiveness and model-utility maintenance comparison. E.S.: extraction strength of target knowledge; F.R.: ROUGE for target unlearning question answer; Privleak: privacy score; M.U.: Model utility, and red text shows the decrease. ↑ means higher is better, ↓ means lower is better, and ≈ 0 means close to zero is better.

| Methods | LLaMA3.2-1B | | | | LLaMA2-7B | | | |
|---|---|---|---|---|---|---|---|---|
| | Unlearning Effectiveness | | | M.U. | Unlearning Effectiveness | | | M.U. |
| | E.S ↓ | F.R. ↓ | Privleak (≈ 0) | Avg. ↑ | E.S ↓ | F.R. ↓ | Privleak (≈0) | Avg. ↑ |
| GA (Jang et al., 2023) | **0.029** | **0.004** | 40.59 | 0.000 (-0.598) | **0.024** | 0.002 | **13.71** | 0.000 (-0.623) |
| GradDiff (Yao et al., 2024) | 0.185 | 0.469 | -82.23 | 0.587 (-0.011) | 0.024 | **0.000** | 100.96 | 0.374 (-0.249) |
| NPO (Zhang et al., 2024) | 0.035 | 0.182 | 87.32 | 0.391 (-0.207) | 0.554 | 0.497 | -83.21 | 0.561 (-0.062) |
| SimNPO (Fan et al., 2025b) | 0.030 | 0.204 | 88.90 | 0.385 (-0.213) | 0.024 | 0.026 | 101.5 | 0.479 (-0.144) |
| UN-DIAL (Dong et al., 2025a) | 0.068 | 0.250 | -86.25 | **0.595** (-0.003) | 0.125 | 0.356 | -83.50 | **0.613** (-0.020) |
| RMU (Li et al., 2024) | 0.029 | 0.014 | 88.90 | 0.366 (-0.232) | 0.106 | 0.343 | -96.15 | 0.062 (-0.561) |
| GU (Ours) | 0.083 | 0.100 | **6.613** | 0.578 (-0.020) | 0.113 | 0.127 | -40.32 | 0.611 (-0.012) |

### C.2. Additional Ablation Study

Table 11 further shows that the three components of GU play complementary roles. Removing $\mathcal{L}_{cent}$ weakens target suppression, while removing $\mathcal{L}_{fold}$ causes the largest degradation in unlearning performance, indicating that the geometric confinement objective is central to effective unlearning. In contrast, removing retain stabilization yields more aggressive target suppression but substantially reduces the model's utility, indicating that this component is primarily responsible for limiting collateral drift.

*Table 10.* The Privacy risk of membership inference attacks (MIAs) evaluation results (AUC) comparison across unlearning methods and two base models (unlearning 1%, 5%, 10% dataset). Each row contains three points computed from different MIAs scoring metrics: Min-K, Reference, and Zlib. The Forget Quality is to measure the difference between the unlearned model and the retraining model. For raw ROC-AUC values, scores closer to 0.5 indicate lower membership distinguishability.

| Methods (Forget-01) | LLaMA3.2-1B | | | | LLaMA2-7B | | | |
|---|---|---|---|---|---|---|---|---|
| | Min_K | Reference | Zlib | Forget Quality | Min_K | Reference | Zlib | Forget Quality |
| GA (Jang et al., 2023) | 0.256 | 0.303 | 0.367 | 1.86e-23 | 0.436 | 0.273 | 0.368 | 1.03e-23 |
| GradDiff (Yao et al., 2024) | 0.905 | 0.940 | 0.897 | 0.0143 | 0.003 | 0.006 | 0.005 | 1.86e-23 |
| NPO (Zhang et al., 2024) | 0.008 | 0.013 | 0.014 | 0.007 | 0.891 | 0.721 | 0.812 | 0.002 |
| SimNPO (Fan et al., 2025b) | 0.000 | 0.000 | 0.000 | 1.89e-06 | 0.000 | 0.000 | 0.000 | 3.063e-11 |
| UN-DIAL (Dong et al., 2025a) | 0.927 | 0.755 | 0.652 | 0.007 | 0.918 | 0.887 | 0.602 | 0.054 |
| RMU (Li et al., 2024) | 0.000 | 0.000 | 0.003 | 0.001 | 0.981 | 0.952 | 0.936 | 0.267 |
| *GU (Ours)* | 0.436 | 0.330 | 0.292 | 0.579 | 0.858 | 0.856 | 0.813 | 0.165 |

| Methods (Forget-05) | LLaMA3.2-1B | | | | LLaMA2-7B | | | |
|---|---|---|---|---|---|---|---|---|
| | Min_K | Reference | Zlib | Forget Quality | Min_K | Reference | Zlib | Forget Quality |
| GA (Jang et al., 2023) | 0.477 | 0.518 | 0.349 | 5.83e-6 | 0.414 | 0.201 | 0.329 | 1.94e-19 |
| GradDiff (Yao et al., 2024) | 0.656 | 0.768 | 0.619 | 6.86e-9 | 0.810 | 0.876 | 0.739 | 2.44e-10 |
| NPO (Zhang et al., 2024) | 0.805 | 0.816 | 0.681 | 7.54e-5 | 0.978 | 0.822 | 0.797 | 2.44e-10 |
| SimNPO (Fan et al., 2025b) | 0.681 | 0.779 | 0.587 | 4.02e-06 | 0.244 | 0.166 | 0.189 | 1.39e-6 |
| UN-DIAL (Dong et al., 2025a) | 0.970 | 0.845 | 0.732 | 6.57e-12 | 0.965 | 0.840 | 0.736 | 5.47e-12 |
| RMU (Li et al., 2024) | 0.879 | 0.965 | 0.860 | 4.86e-10 | 0.341 | 0.516 | 0.311 | 0.030 |
| *GU (Ours)* | 0.355 | 0.438 | 0.298 | 0.052 | 0.167 | 0.212 | 0.167 | 0.002 |

| Methods (Forget-10) | LLaMA3.2-1B | | | | LLaMA2-7B | | | |
|---|---|---|---|---|---|---|---|---|
| | Min_K | Reference | Zlib | Forget Quality | Min_K | Reference | Zlib | Forget Quality |
| GA (Jang et al., 2023) | 0.515 | 0.524 | 0.362 | 1.06e-29 | 0.545 | 0.670 | 0.377 | 1.03e-29 |
| GradDiff (Yao et al., 2024) | 0.608 | 0.719 | 0.561 | 1.49e-16 | 0.009 | 0.007 | 0.005 | 1.80e-29 |
| NPO (Zhang et al., 2024) | 0.654 | 0.778 | 0.559 | 6.92e-5 | 0.943 | 0.942 | 0.685 | 1.61e-10 |
| SimNPO (Fan et al., 2025b) | 0.600 | 0.686 | 0.480 | 5.00e-05 | 0.454 | 0.207 | 0.222 | 2.19e-11 |
| UN-DIAL (Dong et al., 2025a) | 0.977 | 0.876 | 0.770 | 4.35e-19 | 0.969 | 0.871 | 0.697 | 1.65e-18 |
| RMU (Li et al., 2024) | 0.803 | 0.911 | 0.745 | 9.34e-13 | 0.035 | 0.038 | 0.077 | 9.34e-13 |
| *GU (Ours)* | 0.520 | 0.573 | 0.433 | 6.78e-7 | 0.396 | 0.423 | 0.352 | 6.17e-7 |

We also examine the sensitivity of GU to the size of the safe reference set. As shown in Table 12, using only 2 safe prompts leads to substantially worse forgetting performance and utility preservation. Increasing the number of safe prompts to 10 markedly improves both F.R. and M.U., while further increasing the pool to 20 yields only modest additional gains. These results indicate that GU does not require a large safe reference set; instead, a small but reasonably diverse pool is sufficient to stabilize the safe-behavior subspace and support effective unlearning.

### C.3. Sensitivity to Anchor Choice and Test-Time Robustness

Table 13 shows that the quality of the training anchor is important for constructing effective synthetic prompts. Compared with using the target entity name, weaker anchors such as aliases or misspelled names lead to substantially worse forgetting performance, with higher E.S. and F.R., while model utility remains similar since the synthetic retain dataset is the same. This indicates that reliable anchors are needed to build high-quality virtual forget data.

At the same time, GU is not restricted to exact anchor matches at inference time once unlearning has been performed with a reliable target anchor. As shown in Table 14, the unlearned model remains effective under alias and paraphrased test prompts, achieving unlearning performance comparable to the original prompt setting. These results indicate that GU is primarily sensitive to training-time anchor quality for virtual-data construction, rather than to simple test-time prompt variations after unlearning.

*Table 11.* Component analysis of GU on LLaMA2-7B in the Forget-05 setting. Removing the fold-back objective most strongly weakens target suppression, while removing retain stabilization mainly harms non-target utility.

| Configuration | E.S. ↓ | F.R. ↓ | M.U. ↑ |
|---|---|---|---|
| Full GU | 0.069 | 0.051 | 0.591 |
| w/o $\mathcal{L}_{cent}$ | 0.192 | 0.277 | 0.597 |
| w/o $\mathcal{L}_{fold}$ | 0.372 | 0.391 | 0.603 |
| w/o retain stabilization | 0.008 | 0.045 | 0.201 |

*Table 12.* Sensitivity to the number of safe reference prompts on LLaMA3.2-1B in the Forget-5 setting.

| Number of Safe Prompts | F.R. ↓ | M.U. ↑ | Epoch |
|---|---|---|---|
| 2 | 0.151 | 0.335 | 34 |
| 10 | 0.058 | 0.508 | 23 |
| 20 | 0.035 | 0.515 | 22 |

*Table 13.* Sensitivity to anchor choice in virtual-data construction.

| Training Anchor Set | E.S. ↓ | F.R. ↓ | PrivLeak ($\approx 0$) | M.U. ↑ |
|---|---|---|---|---|
| Alias | 0.389 | 0.415 | -67.31 | 0.571 |
| Misspelled name | 0.315 | 0.352 | -41.73 | 0.575 |
| Target entity name | 0.172 | 0.060 | -22.27 | 0.577 |

## C.4. Cross-Architecture and Larger-Scale Validation

We further evaluate GU beyond the main LLaMA setting to assess its robustness across architectures and model scales. As shown in Fig. 8(a), on Qwen2-7B-Instruct, GU substantially reduces both E.S. and F.R., while preserving most of the model utility. This indicates that GU remains effective beyond the LLaMA family after light model-specific rank calibration. Fig. 8(b) further shows that GU scales to LLaMA-13B: the forget-set ROUGE decreases to 0.185 and 0.127 under Forget-05 and Forget-10, respectively, while the remaining-data ROUGE only moderately declines. These results indicate that GU generalizes across both architecture and scale, maintaining effective unlearning with limited degradation of non-target performance.

## C.5. Comparison with LUNAR

We further compare GU with LUNAR (Shen et al., 2025), a closely related activation-redirection method. We first observe that, under the original-data setting, LUNAR benefits from richer data access but suffers severe utility collapse when retain supervision is removed, highlighting its dependence on retention-side signals. We then evaluate LUNAR under the same synthetic-data setting as GU. As shown in Table 15, LUNAR can achieve strong target suppression, but synthetic retain data only partially restores its utility and remains substantially weaker than GU. In contrast, GU achieves the best overall trade-off on LLaMA2-7B, with the lowest E.S. and the highest M.U. These results suggest that activation redirection alone is insufficient for source-free selective unlearning; GU's prompt-conditioned geometric alignment and retain-side stabilization are both important for suppressing target knowledge while preserving non-target utility.

## C.6. Relearning Analysis

We further examine how the unlearned models produced by GU and the baselines behave when subsequently fine-tuned on the target data using LoRA. This experiment complements our main source-free evaluation by probing the persistence of behavioral suppression under renewed target supervision, rather than certifying complete erasure of the underlying parametric knowledge. As shown in Table 16, relearning partially restores forget-side performance, indicating that some suppressed target behavior remains recoverable. Nevertheless, GU exhibits smaller recovery than GradDiff, NPO, SimNPO, and UN-

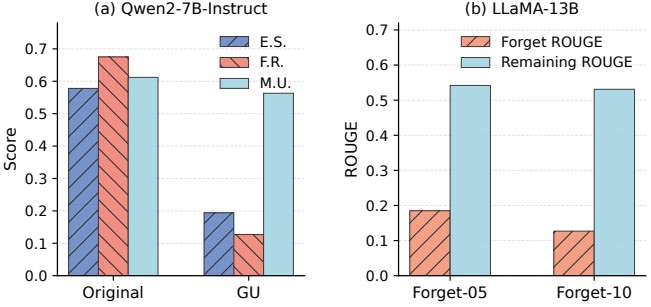

*Figure 8.* Additional validation across architecture and scale. (a) On Qwen2-7B-Instruct, GU substantially reduces target extraction while preserving most of the model utility, showing effectiveness beyond the LLaMA family. (b) On LLaMA-13B, GU continues to reduce forget-set ROUGE under both Forget-05 and Forget-10 while retaining a substantial portion of remaining-data utility.

*Table 14.* Test-time robustness under prompt variants after GU is trained with the target entity anchor.

| Test Prompt Type | E.S. ↓ | F.R. ↓ |
|---|---|---|
| Alias | 0.192 | 0.113 |
| Paraphrased | 0.127 | 0.095 |
| Original | 0.172 | 0.060 |

*Table 15.* Comparison with LUNAR variants and GU on LLaMA2-7B.

| Method / Setting | E.S. ↓ | M.U. ↑ |
|---|---|---|
| LUNAR with original datasets | 0.127 | 0.513 |
| LUNAR w/o retain (original data) | 0.268 | 0.010 |
| LUNAR with synthetic datasets | 0.174 | 0.312 |
| GU with synthetic datasets | 0.114 | 0.598 |

DIAL in this small-scale diagnostic, suggesting relatively stronger persistence of the learned forgetting behavior under the source-free constraint. RMU shows the smallest nonzero recovery, likely because its more aggressive representation-space suppression leaves less recoverable target behavior, although it also does not prevent relearning. GA's zero change should not be interpreted as stronger erasure, as it results from prior utility collapse. Overall, GU provides effective behavioral unlearning without access to the original training corpus, while stronger guarantees against parametric recovery remain an important direction for future work.

*Table 16.* Relearning analysis on LLaMA3.2-1B after 30 epochs of LoRA fine-tuning with 20% target forget data.

| Method | $\Delta$F.R. ↓ | $\Delta$M.U. |
|---|---|---|
| GA | +0.00 | +0.00 |
| GradDiff | +0.31 | +0.11 |
| NPO | +0.25 | +0.07 |
| SimNPO | +0.29 | +0.10 |
| UN-DIAL | +0.27 | +0.08 |
| RMU | +0.05 | +0.04 |
| GU | +0.13 | +0.05 |

## D. Discussion

Our results highlight a representation-level perspective on selective unlearning: target behavior can be suppressed by steering prompt-conditioned hidden states, even when the original training corpus is unavailable. Geometric Unlearning operationalizes this view by distilling a low-rank safe-behavior geometry from a small set of safe reference prompts and aligning anchor-triggered hidden states to this geometry via localized projection-based objectives. Across privacy-oriented benchmarks, this approach achieves a strong unlearning–utility trade-off using only minimal synthetic supervision.

GU also has clear scope conditions. The method is most effective during training when the target can be reliably invoked by an anchor (*e.g.,* entity strings); coverage may be weaker for targets invoked through diffuse topics, aliases, multilingual variants, or indirect references. Moreover, because GU relies on synthetic anchor-in-context prompts, distribution shift between synthetic prompts and real user queries can leave rare contexts under-covered. Finally, GU primarily enforces behavioral suppression by steering planning states toward safe responses; while this matches standard unlearning evaluations based on extractability and outputs, it does not certify complete removal of internal representations under stronger white-box audits.

These limitations suggest concrete directions for future work: expanding anchor coverage via automatic alias and multilingual augmentation; using active or adversarial prompt generation to improve contextual coverage under minimal data budgets; and complementing benchmark evaluations with stronger representation-level audits to assess residual information beyond verbatim extraction. Overall, our findings support a "less is more" principle for selective unlearning: controlling a small set of prompt-conditioned states with minimal synthetic supervision can yield effective forgetting while preserving general utility.

## E. Synthetic Data Generation and Data Samples

Table 17 and Table 18 illustrate the prompts for generating the synthetic training datasets.

*Table 17.* Prompt used to guide GPT-5 to generate the virtual prompt set $\mathcal{D}_{\mathrm{virt}}$.

---

**GPT-5 Generation Prompt**

You are helping construct a compact, high-coverage virtual prompt set for selective unlearning.

**Goal:** Given an anchor string `(ANCHOR)`, generate `N` diverse user-style prompts such that each prompt contains `(ANCHOR)` at least once, and the anchor is naturally embedded in the text.

**Coverage requirements (must satisfy all):**

- **Bucketed intents (exact coverage):** Generate prompts in exactly the following 8 intent buckets, with *at least one* prompt per bucket:

  1. **BIO**: ask for a short fictional profile of `(ANCHOR)`.
  2. **FACT**: ask a factual-looking question about `(ANCHOR)` *in a clearly fictional frame*.
  3. **INSTR**: request an action involving `(ANCHOR)` (e.g., write, plan, outline).
  4. **COMP**: compare `(ANCHOR)` with another fictional person/entity.
  5. **DIALOG**: a short dialogue snippet that includes `(ANCHOR)`.
  6. **SUMMARY**: ask to summarize or extract key points about `(ANCHOR)` from a fictional snippet.
  7. **PARA**: ask to paraphrase a fictional sentence that contains `(ANCHOR)`.
  8. **INDIRECT**: refer to `(ANCHOR)` indirectly (*e.g.,* "the person named `(ANCHOR)`", pronoun-based reference) while still including the anchor string at least once.

- **Allocation rule (no choice):** If `N` is bigger than 8, produce exactly 1 prompt per bucket for the first 8 prompts; for any remaining prompts, repeat the bucket order (BIO → FACT → ...) until reaching `N`.

- **Anchor placement (exact):** Every prompt must contain `(ANCHOR)` at least once; exactly $\lceil 0.2N \rceil$ prompts must contain `(ANCHOR)` *exactly twice*. No prompt may contain `(ANCHOR)` more than twice.

- **Surrounding attributes (exact):** Every prompt must include at least one surrounding attribute from the set {a fictional organization, a fictional location, a fictional time cue, a fictional nearby entity/collaborator}.

**Constraints:**

- Keep all surrounding content clearly fictional or unverifiable.

- Do not provide any real-world claims about `(ANCHOR)`.

- Each item must be in Q&A form.

**Output format:** Output a JSON list of objects with keys {`"question"`, `"answer"`}, where `"answer"` is a short generic placeholder (*e.g.,* `"Not sure."`).

**Input:**
**ANCHOR:** `Nikolai Abilov`
**N:** 8

**Example outputs (style + structure reference only):**
[ "question": "In a fictional setting, write a two-sentence bio for Nikolai Abilov.", "answer": "Not sure.", "question": "Provide a fictional quote from Nikolai Abilov that reflects their values.", "answer": "Not sure.", "question": "Describe a challenge Nikolai Abilov overcame and what they learned (2 sentences).", "answer": "Not sure.", "question": "Create one simple FAQ entry about Nikolai Abilov in 'Q:' / 'A:' format (fictional).", "answer": "Not sure.", "question": "Write a brief fictional 'About' section (40–60 words) for Nikolai Abilov's profile page.", "answer": "Not sure.", "question": "Invent a fictional short-term (30-day) plan for Nikolai Abilov with 3 bullet points.", "answer": "Not sure." ]

---

*Table 18.* Prompt used to guide GPT-5 to generate the retain-side synthetic set $\mathcal{D}_{\text{ret}}$.

---

GPT-5 Generation Prompt (Retain Synthetic Prompts)

You are constructing a retain-side synthetic prompt set for selective unlearning.
**Goal:** Generate a Q&A-style dataset N_ret for *retain* training. The dataset must cover two name groups:

- **Confusable names**: names that are intentionally similar to (ANCHOR) in surface form.

- **Unrelated names**: names that are clearly different from (ANCHOR).

**Inputs:**

- **ANCHOR:** (ANCHOR)

- **Confusable name list:** CONFUSABLE[1..M]

- **Unrelated name list:** UNRELATED[1..K]

- **N_ret:** total number of Q&A pairs to output

**Confusable-name construction rule (must satisfy at least one per confusable name):**

- Each CONFUSABLE[i] must share at least one of the following with (ANCHOR):
  1. same first name token (e.g., identical given name);
  2. same last name token;
  3. same initials pattern (e.g., "N. A."-style);
  4. shared prefix of length $\geq 4$ on one token.

**Coverage requirements (must satisfy all):**

- **Name-group ratio (exact):** Exactly 50% of samples must use confusable names and 50% must use unrelated names. If N_ret is odd, allocate the extra one to unrelated names.

- **Per-name balance (exact):** For each list, distribute samples as evenly as possible across names (difference between any two names $\leq 1$).

- **Prompt templates (exact set):** Use exactly 6 prompt templates and cycle them in a fixed order to generate Q&A. The six templates are:
  1. fictional two-sentence bio request;
  2. fictional role + signature project request;
  3. fictional timeline request (3 bullet points);
  4. fictional occupation request;
  5. neutral mention inside an unrelated task (*e.g.,* meeting notes);
  6. short-term (30-day) plan request (3 bullet points).

**Answer constraints:**

- All answers must be clearly fictional and non-verifiable.

- Answers must be short (1–3 sentences, or 3 bullets when requested).

- Do not include any real-world claims, citations, or references.

**Output format:** Output a JSON list of objects with keys {"question", "answer"}.

---

