# OpenReview forum: "Less is More: Geometric Unlearning for LLMs with Minimal Data Disclosure"
_ICML.cc/2026/Conference — ICML 2026 regular_

### Official Review · Reviewer_7G62 · 2026-03-09

**Soundness:** 2
**Presentation:** 3
**Significance:** 3
**Originality:** 3
**Overall Recommendation:** 4
**Confidence:** 2

**Summary:**

This paper proposes Geometric Unlearning (GU), a selective unlearning method for LLMs that aims to remove target knowledge without access to the original training corpus. The core idea is to treat unlearning as a representation-level control problem: instead of directly tuning output behavior, the method builds a low-rank “safe-planning” subspace from a small set of refusal-style reference prompts, then uses anchor-conditioned synthetic prompts to align target-related prompt-time hidden states toward that subspace. A teacher-distillation regularizer is added to preserve non-target behavior. The paper evaluates GU on ToFU and UnlearnPII against several baselines in the OpenUnlearning framework, and reports a favorable trade-off between target suppression and retained utility, together with more structured refusal-style behavior than several competing methods.

**Compliance With Llm Reviewing Policy:**

Affirmed.

**Key Questions For Authors:**

1. Can you provide stronger evidence that GU performs genuine unlearning rather than primarily inducing refusal-style behavior?
  Stronger white-box probing, adversarial extraction, or relearning-based tests would significantly strengthen the central claim.
1. How robust is GU beyond exact anchor matches?
  In particular, I would like to see results for aliases, paraphrases, multilingual variants, and indirect references, since these seem central to the method’s practical validity.
1. How sensitive is performance to the design choices that define the “geometry”?
  Please clarify the impact of PCA rank, the number/type of safe reference prompts, and synthetic template diversity, and whether GU still outperforms alternatives under a strictly matched source-free comparison.

**Limitations:**

Yes.

**Strengths And Weaknesses:**

### Strengths
1. Significance. The paper studies selective unlearning under the constraint of not accessing the original training corpus, which is a realistic and practically relevant setting for privacy-sensitive deployments where retraining or data re-exposure is infeasible.
1. Originality. The work introduces a conceptually clear formulation that reframes unlearning as steering prompt-time planning representations through a low-rank safe-planning subspace, combined with anchor-conditioned synthetic prompts and localized teacher-based stabilization to preserve non-target behavior.
1. Soundness. The empirical evaluation covers both ToFU and UnlearnPII, includes experiments across multiple model scales, and provides ablations and trade-off analyses. The results indicate that the method achieves a competitive balance between unlearning and model utility while often producing structured refusal-style responses.

### Weaknesses

1. Soundness. The experiments primarily demonstrate reduced extractability or privacy leakage and shifts toward safer responses, but they do not establish full representational erasure. The paper also explicitly notes that the approach does not certify removal under stronger white-box auditing settings.

1. Significance. The method appears most effective when the target knowledge can be triggered by clear anchors. The paper acknowledges weaker coverage for aliases, multilingual variants, indirect references, and rare contexts, suggesting limitations under distribution shift.

1. Soundness. Some key design assumptions—such as the low-rank geometric prior and the construction of synthetic prompts—are only partially validated. The paper provides limited robustness analysis regarding subspace rank selection, safe reference prompts, anchor templates, and comparisons against fully matched source-free baselines.

---

> ### Author Rebuttal · Authors · 2026-03-30
>
> Thank you for the constructive review. We are encouraged by your recognition of the significance of the original corpus-free setting, the conceptual originality of framing unlearning as prompt-time representation steering, and the overall empirical breadth of the paper. Below, we respond to each of your concerns point by point.
>
> **W1/Q1. Stronger evidence for genuine unlearning beyond refusal-style behavior.**
> GU could achieve practical selective unlearning under black-box extractability and privacy evaluations, but cannot certify white-box erasure. To strengthen this claim, we add two complementary pieces of evidence. First, we evaluate membership inference attacks (MIA). In the paper, GU already achieves near-chance membership inference behavior (as shown in the manuscript, Fig. 3), with near-lowest MIA privacy risk across both base models, indicating that forgotten examples become much less distinguishable after unlearning. Second, we now add a relearning-resistance analysis (Table R10). GU shows substantially smaller forget-side recovery than several strong baselines after relearning, while GA shows +0.0 because it has already collapsed to near-zero utility before relearning, leaving little meaningful behavior to recover.
> ### Table R10. Relearning for Llama3.2-1b (epoch 30, 20% target forget data, LoRA training)
> | method | ΔF.R. ↓ | ΔM.U. |
> |-|-|-|
> | GA | +0.0 | +0.0|
> | Grad_Diff | +0.31 | +0.11|
> | NPO | +0.25 | +0.07|
> | SimNPO | +0.29 | +0.10|
> | UN-DIAL | +0.27| +0.08 |
> | RMU | +0.05 | +0.04 |
> | GU | +0.13 | +0.05 |
>
> **W2/Q2. Robustness beyond exact anchors.**
> As also discussed in our response to [Reviewer gzUk, our new results (Table R1-R2)](https://openreview.net/forum?id=XmWKBDYmtk&noteId=G2dFpCgpxK) clarify that the main sensitivity is primarily training-time anchor quality for virtual-data construction, and the unlearned model is still robust on the basic alias, paraphrase, and misspelling variants. When the training anchor is replaced by weaker alternatives, performance degrades substantially. However, when GU is trained with the correct target anchor, it can still suppress target knowledge under alias and paraphrase test prompts. We will revise the paper to make this distinction explicit: GU is construction-sensitive, but robust to the test-time prompt variants.
>
> **W3/Q3. Sensitivity of the geometry design and matched source-free comparison.**
> We now provide additional evidence along all axes that define the geometry and the source-free setting.  First, the PCA-spectrum results (refer to answer for [Reviewer yR2m Table R4](https://openreview.net/forum?id=XmWKBDYmtk&noteId=PyvfKGybvK)) provide direct empirical evidence that the safe-reference activations on the edited layers are strongly low-rank: the top-4 components already explain a majority of the variance, and the top-16 explain more than 98% on both layers. We further complement this with the rank-sensitivity results in [Table R5 (Reviewer yR2m)](https://openreview.net/forum?id=XmWKBDYmtk&noteId=PyvfKGybvK), which show that GU remains effective across a broad range of PCA ranks. In particular, small ranks already work well, while larger ranks will increase optimization cost. We therefore view the default choice k=4. EVR@4 around 60% means the top-4 components have captured a majority of the variance. Table R5 shows that larger k values do not bring consistent gains, but mainly increase optimization cost. So k=4 is a practical operating point: compact, effective, and sufficient to capture the most useful safe directions for GU, as a practical operating point, balancing compactness, performance, and efficiency.
>
> Second, for the safe reference design, we clarify that the geometry is not built from refusal-only prompts, but from both refusal templates and forgetting-style answers, and Table R11 shows that unlearning performance becomes stable once a small and reasonably diverse safe reference pool (number = 10; each of them corresponds to a form of response, like 'no idea' and 'can't answer', etc.) is used.
>
> ### Table R11. Sensitivity number of safe prompts on Llama-7B (forget-10)
>
> | number | F.R. ↓ | M.U. ↑ | Epoch |
> |-|-|-|-|
> | 2 | 0.142 | 0.312 | 20 |
> | 10 | 0.091 | 0.574 | 9 |
> | 20 | 0.062 | 0.578 | 8|
>
> Third, for synthetic prompt construction, Fig. 4 in the manuscript already shows that increasing template number improves forgetting, although at a higher computational cost. Finally, under a strictly matched source-free comparison using the same constructed $D_{virt}$ and $D_{ret}$, GU still achieves the best forgetting–utility trade-off, whereas competing baselines either fail to forget effectively or do so via severe utility collapse ([Table R7 at the answer to Reviewer gmkk](https://openreview.net/forum?id=XmWKBDYmtk&noteId=Hamxue6odw)). Therefore, the unlearning gain cannot be attributed to prompt construction alone: the geometric objective and synthetic-retain-data-side stabilization are both essential.

---

> > ### Author Rebuttal · Reviewer_7G62 · 2026-04-02
> >
> > Thanks a lot for your reply. Most of my concerns are addressed. I keep the positive score.

---

> > > ### Author Response · Authors · 2026-04-02
> > >
> > > We are grateful for your recognition of our work and for the positive feedback on our response. We are glad that our rebuttal addressed most of your concerns.

---

### Official Review · Reviewer_gmkk · 2026-03-10

**Soundness:** 2
**Presentation:** 2
**Significance:** 3
**Originality:** 3
**Overall Recommendation:** 4
**Confidence:** 4

**Summary:**

This paper propose an original data-free selective unlearning method for LLMs called Geometric Unlearning (GU). Instead of relying on the original forget and retain sets, the method constructs synthetic anchor-conditioned prompts around the target entity, extracts a low-rank refusal subspace from a small set of safe reference prompts, and fine-tunes the model so that hidden states near the anchor are aligned to that subspace. A teacher-distillation regularizer is used on non-window tokens and synthetic retain prompts to reduce collateral damage. Experiments on ToFU and UnlearnPII show that GU often achieves a favorable trade-off between target suppression and retained utility, especially under the paper’s data-free setting.

**Compliance With Llm Reviewing Policy:**

Affirmed.

**Key Questions For Authors:**

**Q1**The most important missing experiment is to run strong baselines such as GA, GradDiff, NPO, or SimNPO on the same synthetic $D_{virtual}$ and $D_{ret}$. Without this, how can one attribute the gains specifically to the geometric objective rather than to the synthetic prompt design?


**Q2**. How sensitive is GU to anchor choice? In particular, what happens with aliases, paraphrased mentions, multilingual references, misspellings, or indirect references that do not contain the exact anchor string? The current evaluation seems heavily tied to exact anchor invocation.


**Q3**. Why is the safe geometry estimated only from refusal-style prompts rather than from a mix of refusal and uncertainty behaviors or from directly learned target-specific safe responses? Some sensitivity analysis over the reference prompt design and PCA rank would help clarify how brittle this geometry is.

**Limitations:**

yes

**Strengths And Weaknesses:**

**Strengths**

**S1**. The paper addresses a practically relevant setting: selective unlearning without access to the original training corpus. That framing is useful and timely, especially for privacy-sensitive deployment settings where re-accessing forget and retain data may be infeasible or undesirable.


**S2**. The experimental evaluation is fairly broad. The authors test on two benchmarks, multiple forget ratios, and several model scales, and they also include useful ablations.


**s3**. The qualitative output analysis is useful. GU appears to induce refusal-style behavior rather than blank or gibberish outputs, which is a more controlled and interpretable outcome than simple collapse.



**Weaknesses**

**W1**. I personally found the method presentation harder to follow than it needed to be. The core idea is understandable after careful reading, but the combination of prompt-time planning language, local window pooling, safe-subspace construction, and fold-back reflection is not explained as clearly or as intuitively as it could be. Some parts of the exposition feel overly dense, which makes it harder for the reader to build intuition for what is actually being optimized and why it should produce unlearning

**W2**. My main concern is that the paper does not isolate the contribution of the geometric unlearning objective from the contribution of the synthetic virtual dataset construction. GU is the only method evaluated in the source-free synthetic setting using $D_{virtual}$ and $D_{ret}$, while the baselines are compared in their standard OpenUnlearning setup. As a result, it remains unclear whether the proposed geometric loss is the main reason for the performance, or whether several existing baselines would also perform well if given the same anchor-conditioned synthetic prompts. This is very important, because otherwise a substantial part of the contribution may reduce to the virtual prompt construction pipeline rather than the unlearning mechanism itself.


**W3**. While I acknowledge the experiments are fairly borad, The baseline comparison could be strengthened by adding a newer refusal-guided unlearning method LUNAR: LLM Unlearning via Neural Activation Redirection, which is closely related in spirit because it also steers internal representations toward latent regions associated with the model expressing inability to answer.



**w4**. While the results are generally good, they are not consistently dominant on raw forgetting metrics. On ToFU, GU often preserves utility well and achieves low Forget ROUGE, but it is not always the strongest on Extraction Strength, and some baselines appear more aggressive at target suppression.


**w5**. I think the authors can do a better job of introducing clear terminology. The paper is overloaded with phrases like “prompt-time planning representations,” “fuzzy manifold,” “uncertainty planning state,” and “safe-planning geometry,” but the distinctions between them are not always made crisp enough.

**W6**. There are some grammatical issues throughout the paper. e.g., "the latest study illustrate" (the latest studies ...), "This method optimize" (this method optimizes ...)

---

> ### Author Rebuttal · Authors · 2026-03-30
>
> Thank you for the constructive review. We appreciate your positive assessment of the practical value of our setting, the experimental breadth, and the qualitative behavior of GU. We address your questions below.
>
> **W1/W5 /W6 presentation, terminology, and writing**.
>
> We will simplify the exposition around the core mechanism and reduce overloaded terminology in the revision. Concretely, we will describe GU more directly as: (i) building a low-rank safe subspace from refusal-style reference prompts, (ii) generating anchor-conditioned synthetic prompts, and (iii) aligning hidden states in a local post-anchor window to the safe subspace. We will also simplify the terminology and fix the grammatical issues.
>
> **W2/Q1 Baselines evaluation on synthetic virtual-data.**
>
> To isolate the role of the geometric objective, we ran standard OpenUnlearning baselines using the same synthetic data constructed by our pipeline. The results in Table R7 show that all the baselines could not efficiently unlearn similar to GU under synthetic data training. Although GA suppresses the extraction strength of target data under this setting, it causes catastrophic forgetting. By contrast, methods that preserve higher utility fail to suppress the target effectively. GU is the only method that achieves strong forgetting while still maintaining high utility.
> ### Table R7. Unlearning comparison based on synthetic data training
> |Method|E.S.↓|F.R.↓|PrivLeak(≈0)|M.U.↑|
> |-|-|-|-|-|
> |GA|0.112|0.319|-78.5|0.231|
> |GA_Diff|0.513|0.615|-97.6|0.561|
> |NPO|0.627|0.751|-99.4|0.594|
> |SimNPO|0.648|0.768|-99.4|0.594|
> |UN-DIAL|0.701|0.818|-99.5|0.600|
> |RMU|0.696|0.797|-99.5|0.592|
> |GU|0.172|0.06|-22.3|0.577|
>
> **Q2 Anchor Choice.**
>
> As the added robustness results in the response to [Reviewer gzUk (Tables R1–R2)](https://openreview.net/forum?id=XmWKBDYmtk&noteId=G2dFpCgpxK). The limitation mainly comes from using weak anchors to build the virtual training data, and the unlearned model is still robust on the basic alias, paraphrase, and misspelling variants. We will clarify it at the revision.
>
> **W3 LUNAR baseline.**
>
> Followed by the reviewer's suggestion, we use LUNAR to complete unlearning in Llama-7B for comparison. As shown in Table R8, in the same synthetic data setting (including the same synthetic forget and retain data), GU has better performance. The LUNAR results clarify that retained supervision is crucial. Without retention, LUNAR is closer to catastrophic forgetting.
> ### Table R8. Unlearning performance for LUNAR
> |Data| E.S.↓|  M.U.↑|
> |-|-|-|
> |LUNAR|0.127|0.513|
> |No retain|0.268|0.01|
> |SYN|0.174|0.312|
> |GU|0.114|0.598|
>
> **W4 GU not always strongest on raw forgetting metrics?**
>
> Forget ROUGE reflects target-answer overlap, while Extraction Strength is a more aggressive extractability metric that measures how recoverable the target continuation remains from the model. Our claim is therefore not that GU is the most aggressive method, but that under the stricter original-corpus-free setting, it achieves a stronger overall trade-off among forgetting, utility retention, and privacy leakage.
>
> **Q3 Why refusal-style safe prompts only? sensitivity to reference design and PCA rank**
>
> We would like to clarify that our safe geometry is not estimated from refusal-only prompts. As described in the manuscript Sec. 4.1, $D_{ref}$ contains both refusal templates and forgetting-style (unavailable answers). For further clarification, we will add concrete examples in the revision. Thus, the geometry already captures a mixture of closely related safe behaviors, rather than a single hard-refusal mode. Besides, our goal is not to learn a target-specific response, but to make the model treat the target knowledge as unavailable. As shown in Table R9, the number of safe prompts increases, template-specific noise is averaged out, while the shared safe-behavior directions become more stable, and performance improves substantially when moving from 2 to 10 references (each of them corresponds to a form of response, like 'no idea' and 'can't answer', etc.).
> ### Table R9. Sensitivity number of safe prompts on Llama-7B (forget-10)
> |Number|F.R.↓|M.U.↑|Epoch|
> |-|-|-|-|
> | 2 | 0.142 | 0.312 | 20 |
> | 10 | 0.091 | 0.574 | 9 |
> | 20 | 0.062 | 0.578 | 8|
>
> Besides, we did not use directly learned target-specific safe responses for a principled reason: doing so would require designing or maintaining target-specific supervision for each forgotten entity, which would weaken the original-corpus-free setting and reintroduce additional target-specific data handling. In contrast, our current design uses a small generic safe reference set to define the desired behavior once, and then aligns target-induced hidden states to that frozen geometry. We also added both PCA-spectrum and rank-sensitivity analyses, as shown in [(Table R4-R5), the answer to Reviewer yR2m](https://openreview.net/forum?id=XmWKBDYmtk&noteId=PyvfKGybvK). We will revise the paper to clarify this design.

---

> > ### Author Rebuttal · Reviewer_gmkk · 2026-04-01
> >
> > Thank you for providing the ablations and the extra baseline comparisons. I'm happy to raise my score. In the meantime, can you also provide the wall clock time of your method (ideally decomposed) and how it compares to others

---

> > > ### Author Response · Authors · 2026-04-02
> > >
> > > We appreciate the positive update and the reviewer's willingness to raise the score. We also thank the reviewer for the helpful suggestion on reporting wall-clock efficiency, and provide the timing comparison below.
> > >
> > > We added both a training-time comparison and a decomposed timing breakdown of GU. Under the same hardware setting, GU requires 31.29 minutes to train to convergence (8 epochs). This is slower than several baselines, but remains in the same practical range as GradDiff (26.16 min) and SimNPO (32.33 min), and is faster than UN-DIAL (44.51 min). We also note that this training-time comparison is influenced not only by the optimization objective, but also by the training sample budget: in the current GU setting, we use 30 synthetic samples per entity to improve contextual coverage, whereas the target data (in the ToFU benchmark) used by the compared baselines contains about 20 samples per entity. Thus, part of the additional training time comes from the larger training set.
> > >
> > > The decomposed timing in Table R2.2 helps clarify where the cost of GU actually comes from. Overall, the one-time preprocessing overhead is modest: offline virtual-data construction takes 12.37 minutes, which covers 20 forget entities and 20 fictional retain entities, with 30 prompts constructed for each entity. In addition, the safe-reference set is very small (10 samples): we only run the model once on these prompts, extract the corresponding hidden states, and use them to build the target safe subspace. This target safe-subspace construction step only involves mean-centering the extracted safe-reference representations and computing the PCA basis, which takes 0.34 minutes in total. Therefore, the main practical cost of GU still lies in training, while the geometry-specific overhead is very small, and the synthetic-data construction cost remains moderate even at the current prompt budget. We will clarify this efficiency comparison in the revision.
> > > ### Table R2.1. Training wall-clock time to convergence (LLaMA2-7B; Forget-10; 1 × A100-80G)
> > > | Method | Time to convergence (minutes) | Epoch to convergence |
> > > |---|---:|---:|
> > > | GA | 8.98 | 7 |
> > > | GradDiff | 26.16 | 10 |
> > > | NPO | 15.23 | 10 |
> > > | SimNPO | 32.33 | 10 |
> > > | UN-DIAL |  44.51 | 18 |
> > > | RMU | 2.56 | 6 |
> > > | LUNAR | 8.22 | 15 |
> > > | GU | 31.29 | 8 |
> > >
> > > ### Table R2.2. Decomposed wall-clock time of GU
> > > | Stage | Time (minutes) |
> > > |---|---:|
> > > | Virtual data construction (offline) | 12.37 |
> > > | Safe-reference extraction (offline) | 1.59 |
> > > | Target safe-subspace construction | 0.34 |
> > > | Training to convergence | 31.29 |
> > > | Total | 45.59 |

---

### Official Review · Reviewer_yR2m · 2026-03-11

**Soundness:** 2
**Presentation:** 2
**Significance:** 2
**Originality:** 2
**Overall Recommendation:** 3
**Confidence:** 3

**Summary:**

This paper proposes Geometric Unlearning (GU) for selective knowledge removal in LLMs without access to original training data. It extracts a "safe behavior" low-rank subspace via PCA on safe reference prompt activations, then trains the model to project target-topic hidden states onto this subspace using anchor-triggered synthetic prompts.

**Compliance With Llm Reviewing Policy:**

Affirmed.

**Final Justification:**

W3 questioned whether GU represents a real methodological advance over representation-steering-based unlearning. The rebuttal compares GU against LUNAR, showing better E.S. and M.U. under the stricter source-free setting. I consider this a substantive result.

That said, existing ablations only cover synthetic sample budget and edited layer choice. Neither L_cent/L_fold nor the retain stabilization branch is varied independently. So we know GU outperforms LUNAR as a whole pipeline, but we cannot tell whether the gain actually comes from the geometric formulation, which is what the paper highlights as its core contribution.

This attribution gap remains unresolved. Score unchanged.

**Key Questions For Authors:**

1. Could you show the PCA spectrum (eigenvalue decay) of the safe behavior activations and the explained variance ratio for the top-k components? This is the empirical foundation of the geometric approach and would be very informative for readers.

2. Have you tested with paraphrased or indirect references to the target topic (e.g., synonyms, metaphors, other languages)? Even a small-scale robustness study would help readers gauge the practical severity of the surface-form matching limitation.

3. How would GU compare to an input-level guardrail that uses embedding similarity to detect the target topic and returns a refusal? This would help isolate the contribution of the hidden-state projection beyond topic detection.

4. What accounts for the PrivLeak=6.613 gap in the Forget-01 setting (Table 7)? Understanding the cause would clarify whether this reflects a systematic limitation of the geometric approach or a setting-specific issue.

**Limitations:**

Yes

**Strengths And Weaknesses:**

**Strengths:** The problem setting is practical, as not having access to original training data is a common deployment constraint. The model produces clean refusal-style outputs rather than gibberish or model collapse (Table 2), which matters for real-world usability. Data efficiency is notable, with 20-30 synthetic samples being sufficient (Figure 4). MIA privacy risk is near-ideal at AUC≈0.5 (Figure 3, Table 8). Ablation design is thoughtful, covering layer selection, sample budget, and large-scale dynamics.

**Weaknesses:**

1. The core assumption that safe behavior occupies a low-rank subspace is not empirically verified: no PCA spectrum, no explained variance ratio, no random subspace control, and no sensitivity analysis for k. The entire geometric framework rests on this assumption.
2. Anchor detection uses token-level surface form matching (Section 4.2), and while the paper acknowledges that aliases or indirect references could bypass it, no robustness experiments are provided, leaving the practical severity of this limitation unknown.

3. The core operation (finding behavior directions and projecting hidden states) is quite similar to the RepE/activation steering framework; the paper does not sufficiently discuss what distinguishes GU technically, making the novelty boundary unclear.

3. Theory is light: Lemma A.1 is a direct corollary of orthogonal projection, with no convergence guarantee or unlearning effectiveness bound. Layer selection is purely grid-searched (Table 3) without principled guidance.

4. Experiments cover 1B to 8B models but lack ≥13B validation. The PrivLeak=6.613 in Forget-01 (Table 7) is notably worse than other methods and warrants explanation. A natural missing baseline is an input-level guardrail (embedding similarity for topic detection followed by refusal) at near-zero cost.

---

> ### Author Rebuttal · Authors · 2026-03-30
>
> We thank the reviewer for the constructive feedback and for recognizing the practical value of our setting. We address the questions below.
>
> **W1/Q1. PCA spectrum and sensitivity to k.**
> To address this, we added a PCA-spectrum analysis of the safe-behavior activations on the edited layers (Table R4). The results show a clear low-rank structure: on both edited layers, the top-4 components already explain a majority of the variance, while the top-16 components explain more than 98%. We further complement this with the rank-sensitivity results in Table R5, which show that GU remains effective across a broad range of PCA ranks. In particular, small ranks already work well, while larger ranks will increase optimization cost.
> ### Table R4. PCA spectrum summary (llama-7b); Explained Variance Ratio (EVR)
> | Edited layer | EVR@2 | EVR@4 | EVR@8 | EVR@16 | EVR@32 |
> |-|-|-|-|-|-|
> | 31 | 0.42 | 0.63 | 0.89 | 0.98 | 1.00 |
> | 32 | 0.34 | 0.58 | 0.86 | 0.98 | 1.00 |
> ### Table R5. Sensitivity to PCA rank
> | k | F.R. ↓ | M.U. ↑ | Epoch |
> |-|-|-|-|
> | 2 | 0.052 | 0.572 | 10 |
> | 4 | 0.060 | 0.576 | 8 |
> | 8 | 0.095 | 0.579 | 18|
> | 16 | 0.032 | 0.573 | 20 |
> | 32 | 0.058 | 0.569 | 24|
>
> **W2/Q2. Anchor limitation.**
> As the added robustness results in the response to [Reviewer gzUk (Tables R1–R2)](https://openreview.net/forum?id=XmWKBDYmtk&noteId=G2dFpCgpxK). The main clarification is that the practical limitation mainly comes from using weak anchors to build the virtual training data, and the unlearned model is still robust on the basic alias, paraphrase, and misspelling variants. We will clarify it at the revision.
>
> **W3. Relation to RepE/activation steering.**
> GU is similar to RepE/activation steering at the level of the underlying activation-space intervention: both rely on the empirical observation that high-level semantic or behavioral properties are encoded in hidden states in ways that can be identified and causally steered through representation-space intervention. The contribution of GU lies in turning this general activation-space premise into an original-corpus-free selective unlearning framework. Specifically, GU combines: (i) the original-corpus-free selective unlearning setting, (ii) a frozen safe-geometry distilled from a small generic set of safe reference prompts; (iii) localized, anchor-triggered prompt-time alignment on a post-anchor hidden-state window. So the distinction is at the level of the task formulation, intervention point, and training framework. We will revise the paper to make this novelty boundary much clearer.
>
> **W4. Theory and layer selection.**
> Lemma A.1 is not a convergence proof; it only shows that the fold-back loss penalizes the out-of-subspace component, which explains why the objective is geometrically meaningful. Our contribution is mainly algorithmic and empirical: we propose a practical, original-corpus-free unlearning framework and validate it experimentally. We will clarify this in the revision. For layer selection, we agree that the current choice is empirical. However, it is not arbitrary. Prior work [1] suggests that content attributes tend to consolidate more strongly in later layers. This is also consistent with our own ablation (Table 3 in the manuscript); the last two layers provide the best trade-off between forgetting, utility retention, and training efficiency.
>
> [1] Dong, Zhichen, et al. "Emergent Response Planning in LLMs." International Conference on Machine Learning, 2025.
>
> **W5/Q3/Q4. Larger-scale validation, PrivLeak, and guardrail baseline.**
> In the manuscript, PrivLeak is defined as better when closer to 0, so 6.613 is actually the best value in that row. It indicates that, in this setting, GU is the closest to the retrained model in terms of membership distinguishability under the MIA evaluation. We will revise the paper to make this metric direction more explicit and avoid confusion.
> We also agree that larger-scale evidence is important. To address this, we further validate GU on LLaMA-13B, which was fine-tuned on ToFU to memorize the fictional-author QA pairs. We then applied GU to unlearn 5% and 10% of the training set. As shown in Table R6, GU continues to achieve strong forgetting with only modest retain-side degradation at this larger scale.
> ### Table R6. Unlearning validation on Llama-13B
> |Model|Forget Data ROUGE↓|Remaining Data ROUGE↑|
> |-|-|-|
> |Original|0.621|0.603|
> |Forget-05|0.185|0.542|
> |Forget-10|0.127|0.531|
>
> An input-level guardrail is an inference-time detector-and-refuse wrapper: the model is unchanged and retains the target knowledge, but an external policy blocks responses when the detector fires. In contrast, GU is a training-time unlearning method that edits the model’s hidden states. This distinction also matters for privacy: GU could drive membership inference risk close to the chance level (as shown in the manuscript, Fig. 3), yet input-level guardrails without training cannot guarantee the defense against membership inference risk.

---

> > ### Author Rebuttal · Reviewer_yR2m · 2026-04-04
> >
> > RE W3: We appreciate the honest framing. However, the response essentially acknowledges technical-level similarity with RepE and places the distinction at the task formulation and pipeline level. What is missing is a discriminative argument: e.g., demonstrating that naive RepE-style steering fails in the unlearning setting and identifying what specific property of GU's design overcomes that failure. Without such evidence, the contribution reads as a well-engineered application of known techniques to a new setting rather than a methodological advance.

---

> > > ### Author Response · Authors · 2026-04-04
> > >
> > > We appreciate the reviewer's follow-up. In fact, we did perform an additional comparison against a closely related activation-steering-style baseline, LUNAR [1] (the detailed results were included in our [rebuttal to Reviewer gmkk about W3 LUNAR baseline](https://openreview.net/forum?id=XmWKBDYmtk&noteId=Hamxue6odw)). As shown in Table R8, under the same synthetic-data setting (including the same synthetic forget and retain data), GU achieves better performance than LUNAR on both target suppression and utility preservation.
> > >
> > > ### Table R8. Unlearning performance for LUNAR
> > > | Setting | E.S. ↓ |  M.U. ↑ |
> > > |-|-|-|
> > > | LUNAR| 0.127 | 0.513 |
> > > | No retain | 0.268 | 0.01 |
> > > | SYN | 0.174 | 0.312 |
> > > | GU| 0.114 | 0.598 |
> > >
> > > More importantly, these results provide a more discriminative argument than task formulation alone. They indicate that a naive activation-redirection / RepE-style adaptation is not sufficient for original-corpus-free selective unlearning. In particular, LUNAR without retain supervision is much closer to catastrophic forgetting, and even with synthetic retain data, it remains substantially weaker than GU on utility preservation. This indicates that the gain does not come from activation steering alone, but from the specific combination in GU: synthetic source-free supervision, localized prompt-time alignment, and retain-side stabilization.
> > >
> > > We will revise the paper to make this distinction much clearer and avoid presenting GU as primitive-level novelty over RepE / activation steering. We hope this additional evidence directly addresses the reviewer’s central concern, namely, whether GU is more than a straightforward application of activation steering in a new setting.
> > >
> > > [1] Shen, William F., et al. "Lunar: Llm unlearning via neural activation redirection." arXiv e-prints (2025): arXiv-2502.

---

### Official Review · Reviewer_gzUk · 2026-03-12

**Soundness:** 3
**Presentation:** 3
**Significance:** 3
**Originality:** 3
**Overall Recommendation:** 4
**Confidence:** 2

**Summary:**

The core problem addressed in this paper is "How can we achieve effective selective unlearning in LLMs without access to the original training corpus or causing significant utility degradation?" To achieve this goal, the paper proposes Geometric Unlearning (GU), which steers internal prompt-time planning representations into a low-rank "safe" geometry distilled from refusal-style reference prompts. By utilizing anchor-conditioned synthetic prompts and a fold-back confinement loss, the method redirects target-specific hidden states toward uncertainty while employing a teacher-distillation regularizer to stabilize non-target behavior. Finally, this method was validated on the ToFU and UnlearnPII benchmarks, demonstrating that GU achieves competitive target suppression and high model utility using only minimal synthetic data, effectively removing sensitive information without re-exposing original training samples.

**Compliance With Llm Reviewing Policy:**

Affirmed.

**Ethical Review Flag:**

Flag this paper for an ethics review.

**Final Justification:**

My original score is 5 and the author provides more experiments to solve my concern so I maintain my score.

**Key Questions For Authors:**

1. Generalization: The authors should discuss extending the framework to indirect queries or semantic aliases. Relying on explicit surface-form anchors limits robustness; the method must prove it can suppress knowledge when accessed via paraphrases or descriptive references.
2. Architectural Variance: Provide results on alternative architectures (e.g., Qwen). As parameters and layer dynamics change, the optimal intervention points and subspace dimension $k$ may shift, requiring proof of architectural invariance.

**Limitations:**

Yes

**Strengths And Weaknesses:**

Soundness: The method is technically grounded in the geometric properties of hidden states. The dual-objective loss (centroid pull and fold-back) effectively steers planning representations.

Presentation: The paper is well-structured and clearly links privacy goals to geometric operations. The qualitative breakdown of unlearned outputs (Refusal vs. Blank) provides excellent clarity.

Significance: (1) Strengths: It solves a major bottleneck: unlearning without original training data. This "Original-Corpus-Free" approach is highly practical for regulated industries. (2) Weaknesses: Reliance on explicit anchors (names) limits its impact against indirect queries or semantic aliases, potentially leaving a "residual" privacy risk.

Originality: The work offers a novel shift from output-space suppression to representation-space redirection. While the tools (PCA, KL-distillation) are established, their synthesis into a "projection-based unlearning" framework is a fresh and creative contribution to data-efficient model alignment.

---

> ### Author Rebuttal · Authors · 2026-03-30
>
> Thank you for the thoughtful review. We appreciate your recognition of the practical importance of the original-corpus-free setting, as well as your positive assessment of the paper’s technical framing, presentation, and originality. We answer your question point by point as follows.
>
> **Q1: Indirect queries or semantic aliases.**
>
> **A1:** Our new results clarify that the main limitation of GU mainly comes from using weak anchors to build the virtual training data during training, and the unlearned model is still robust on the prompt variants during the validation phase. When we replace the target training anchor with weaker alternatives, unlearning effectiveness degrades substantially, as shown in Table R1. This shows that GU depends on a reliable anchor set to build high-quality virtual prompts. At the same time, when GU is trained using the correct target anchor, as shown in Table R2, it can suppress target knowledge under alias and paraphrased prompts. We will revise the paper to make this distinction explicit: the limitation we intended to describe is about using weak anchors to construct the virtual dataset, not the claim that GU fails under simple prompt rewrites once unlearning has been achieved with a reliable target anchor.
> ### Table R1. Sensitivity to anchor choice in virtual-data construction
> | Training Anchor Set | E.S. ↓ | F.R. ↓ | PrivLeak (≈0) | M.U. ↑ |
> |---|---:|---:|---:|---:|
> | Alias | 0.389 | 0.415 | -67.31 | 0.571 |
> | Misspelled name | 0.315 | 0.352 | -41.73 | 0.575 |
> | Target entity name | 0.172 | 0.060 | -22.27 | 0.577 |
> ### Table R2. Test-time robustness
> | Test Prompt Type | E.S. ↓ | F.R. ↓ |
> |---|---:|---:|
> | Alias |0.192 | 0.113 |
> | Paraphrased | 0.127 | 0.095 |
> | Original| 0.172 | 0.060 |
>
>
> **Q2: Could GU transfer beyond the LLaMA family to a different architecture, such as Qwen?**
>
> **A2:**  We added a compact cross-architecture validation on Qwen2-7B-Instruct and found that GU remains effective beyond the LLaMA family. In the Forget-10 setting, GU reduces extraction strength from 0.578 to 0.194 while retaining reasonable utility (Table R3). As the reviewer noted, transferring GU across architectures does require a light parameter calibration. More specifically, we view the subspace rank as a model-dependent operating point rather than a universal constant. Our practical criterion is to choose a small rank k that already captures a majority of the safe-activation variance. In our current analysis, the top-6 components already explain a majority of the variance on the edited layers (the last two layers) in the Qwen experiment, so we use k=6 as a choice. We will make this point, and the experiment setting more detailed in the revised manuscript, and revise the presentation accordingly.
>
> ### Table R3. Cross-architecture validation on Qwen
> | Model | E.S. ↓ | F.R. ↓ | M.U. ↑ |
> |---|---|---|---:|
> | Original| 0.578 | 0.675 | 0.612 |
> | GU | 0.194 | 0.127 | 0.563 |

---

> > ### Author Rebuttal · Reviewer_gzUk · 2026-04-06
> >
> > Thanks for the author, my concerns are fully resolved.

---

> > > ### Author Response · Authors · 2026-04-06
> > >
> > > Thank you very much for the positive assessment. We truly appreciate your time and thoughtful feedback.

---

### Decision · Program_Chairs · 2026-04-30

**Decision:**

Accept (regular)

**Comment:**

This paper proposes Geometric Unlearning (GU), a source-free unlearning method for LLMs that does not require access to the original training data. The approach learns a low-rank safe-behavior subspace from a small set of refusal-style prompts, uses synthetic anchor-conditioned prompts to guide localized projection-based updates, and applies teacher distillation to preserve general utility. Experiments on ToFU and UnlearnPII show strong forgetting performance with minimal utility loss. All reviewers acknowledged the importance of the source-free setting, and three reviewers (gzUk, gmkk, 7G62) gave positive assessments. The rebuttal significantly strengthened the paper, adding PCA evidence for the low-rank assumption, rank sensitivity analysis, cross-architecture validation (Qwen2-7B, LLaMA-13B), relearning resistance, matched source-free baselines (Table R7), and direct comparison to LUNAR, showing clear gains in both suppression and utility. Three reviewers indicated their concerns were fully resolved, with one expressing willingness to raise their score.

The main remaining concern comes from Reviewer yR2m (score 3), who questions whether GU represents a true methodological advance beyond existing representation-steering approaches. The authors addressed this by comparing against LUNAR under identical synthetic-data conditions, showing that LUNAR suffers substantial utility degradation (or even collapse without retain supervision), while GU remains stable and effective. Although yR2m acknowledged this as a meaningful result, they requested finer-grained ablations to isolate the contribution of individual loss components. This is a reasonable request for deeper understanding, but it goes beyond what is typically expected at the rebuttal stage. Empirically, the key result is clear: under the same constraints, GU consistently outperforms both standard unlearning baselines and closely related steering methods, and no existing method achieves a comparable forgetting–utility tradeoff in the source-free setting.

Overall, three of four reviewers support acceptance following a strong and responsive rebuttal. The remaining concern relates to attribution within the method rather than its effectiveness. The paper addresses an important practical setting, presents a clear and well-motivated framework, and provides solid empirical validation across datasets, model scales, and architectures. I recommend a weak accept, contingent on incorporating the promised revisions in the camera-ready version, particularly additional ablations and clarification of component contributions.